# Spatio-temporally separated cortical flows and spindle geometry establish physical asymmetry in fly neural stem cells

Chantal Roubinet [1,3], Anna Tsankova[1,4], Tri Thanh Pham[1,2], Arnaud Monnard[1,2], Emmanuel Caussinus[1,5], Markus Affolter[1] & Clemens Cabernard [1,2]

Asymmetric cell division, creating sibling cells with distinct developmental potentials, can be manifested in sibling cell size asymmetry. This form of physical asymmetry occurs in several metazoan cells, but the underlying mechanisms and function are incompletely understood. Here we use *Drosophila* neural stem cells to elucidate the mechanisms involved in physical asymmetry establishment. We show that Myosin relocalizes to the cleavage furrow via two distinct cortical Myosin flows: at anaphase onset, a polarity induced, basally directed Myosin flow clears Myosin from the apical cortex. Subsequently, mitotic spindle cues establish a Myosin gradient at the lateral neuroblast cortex, necessary to trigger an apically directed flow, removing Actomyosin from the basal cortex. On the basis of the data presented here, we propose that spatiotemporally controlled Myosin flows in conjunction with spindle positioning and spindle asymmetry are key determinants for correct cleavage furrow placement and cortical expansion, thereby establishing physical asymmetry.

[1] Biozentrum, University of Basel, Klingelbergstrasse 50-70, CH-4056 Basel, Switzerland. [2] Department of Biology, University of Washington, 24 Kincaid Hall, Seattle, WA 98195, USA. [3]Present address: MRC Laboratory for Molecular Cell Biology, University College London, Gower Street, London WC1E 6BT, UK. [4]Present address: Streuli Pharma AG, Bahnhofstrasse 7, CH-8730 Uznach, Switzerland. [5]Present address: Institute of Molecular Life Sciences, University of Zurich, Winterthurerstrasse 190, CH-8057 Zurich, Switzerland. Anna Tsankova and Tri Thanh Pham contributed equally to this work. Correspondence and requests for materials should be addressed to C.C. (email: ccabern@uw.edu)

Asymmetric cell division is an evolutionary conserved mechanism to create sister cells with divergent fate[1]. One manifestation of asymmetric cell division is the difference in sibling cell size and occurs in various cell types and organisms[2, 3]. Several mechanisms underlying the generation of physical asymmetry have been proposed but how they are spatiotemporally coordinated and molecularly controlled is incompletely understood[4]. Controlled cleavage furrow positioning can generate sibling cell size asymmetry by assembling an actomyosin-containing contractile ring at the correct position underneath the cell membrane. In most metazoan cells, the positional cues regulating ring positioning and assembly originate from the mitotic spindle in the form of the conserved Centralspindlin complex, composed of the mitotic kinesin-like protein 1 (MKLP1) (Pavarotti; Pav in *Drosophila*; Zen-4 in *Caenorrhabditi elegans*) and MgcRacGAP (Tumbleweed; Tum in *Drosophila*; CYK-4 in *C. elegans*)[5–7]. Centralspindlin's localization to the central spindle is controlled by the chromosomal passenger complex (CPC), consisting of Aurora B kinase, the inner centromere protein (INCENP), Survivin and Borealin[8]. It has been proposed that Pav travels along stable microtubules, delivering Tum to the cell equator where it activates the RhoGEF ECT2 (Pebble; Pbl in *Drosophila*; LET-21 in *C. elegans*)[9–11]. Equatorial localization of Pbl induces the activation of the small GTPase RhoA (Rho1 in *Drosophila*), promoting Actin polymerization and Myosin activation, resulting in the formation of the actomyosin-containing contractile ring[5, 6].

This generalized model can explain equatorial Non-muscle Myosin II (Myosin, hereafter) localization in a number of cell types. However, cell type-specific variations, highlighting fundamental mechanistic differences in Myosin dynamics, also exist. For instance, in Sea urchins, phosphorylated Myosin is localized on the cell cortex until metaphase but subsequently disappears from the entire cortex before reappearing in a confined spindle-induced band at the equatorial furrow[10]. In *Drosophila* neuroblasts, the neural stem cells in the developing fly brain, Myosin remains at the cell cortex throughout mitosis but the polarity proteins Discs large 1 (Dlg1; Dlg in vertebrates) and Partner of Inscuteable (Pins; LGN/AGS3) are used to transform Myosin from a uniform cortical distribution to an asymmetric localization before it enriches at the forming cleavage furrow[12]. Spindle-independent furrow positioning mechanisms are not confined to the neuroblast system but have also been reported in other organisms and cell types[13–17].

Myosin localization also influences the stability and dynamic behavior of the cell cortex. For instance, asymmetric Myosin localization regulates biased cortical expansion, shifting the cleavage furrow towards one cell pole, thereby generating unequal sized sibling cells and thus physical asymmetry[13, 18]. However, how Myosin dynamics and activity are spatiotemporally regulated to ensure the correct establishment of physical asymmetry, remains unclear.

Here we use photoconversion, live cell imaging, laser cutting and nanobody experiments in the *Drosophila* neuroblast system to specifically investigate the molecular mechanisms underlying sibling cell size asymmetry. We show that Myosin relocalizes to the cleavage furrow via two distinct cortical Myosin flows: a polarity induced, basally directed Myosin flow, causing Myosin to clear on the apical cortex at anaphase onset. Subsequently, mitotic spindle cues establish a Myosin gradient at the lateral neuroblast cortex, necessary to trigger an apically directed flow, removing Myosin from the basal cortex. On the basis of the data presented here, we propose that both spatiotemporally controlled Myosin flows in conjunction with spindle positioning and spindle asymmetry are key determinants for correct cleavage furrow placement and cortical expansion and thus the establishment of physical asymmetry.

## Results

**Cell cycle and polarity cues regulate Myosin dynamics.** To learn how Myosin dynamics contributes towards sibling cell size asymmetry, we used live cell imaging and measured the relocalization dynamics of Non-muscle Myosin II (visualized with Sqh::GFP[19]; Myosin (Myo), hereafter) together with the cell cycle marker His2A::mRFP in wild-type fly neuroblasts. We confirmed that Myosin was localized almost uniformly around the cortex by late metaphase[12, 18, 20]. Approximately 20 s after anaphase onset, Myosin first disappeared from the apical cortex and ~ 80 s later from the basal cortex, resulting in a ~ 1-minute delay between apical and basal Myosin depletion. Myosin also accumulated at the basally shifted cleavage furrow (Fig. 1a–c). Live cell imaging with high temporal resolution revealed that apical relocalization preceded Myosin enrichment at the lateral cortex—the future furrow position—by 25 s (+/− 8 s; n = 17. "+/−" refers to standard deviation (s.d); "n" refers to number of measured cells). Myosin enrichment at the lateral cortex also preceded basal Myosin clearing by 30 s (+/−17 s; n = 17). Myosin then continued to enrich at the prospective furrow position once basal clearing was initiated (Fig. 1d).

This stereotypic Myosin relocalization sequence (summarized in Supplementary Fig. 1a) depended on both cell cycle and polarity cues. Partial inhibition of Cyclin dependent kinase 1 (Cdk1) has been shown to be sufficient to initiate cytokinesis[21]. We used Flavopiridol to partially inhibit Cdk1 in wild-type fly neuroblasts and measured the time between nuclear envelope breakdown and apical clearing, basal clearing and Myosin enrichment in the furrow region. We found that apical and basal Myosin relocalization occurred earlier compared to wild-type neuroblasts. Similarly, Myosin enrichment at the future cleavage furrow also occurred prematurely (Fig. 1e–g).

Previously, we showed that the polarity proteins Pins and Dlg are necessary for the correct localization of Myosin[12, 18, 20]. Since neuroblast polarity is connected with the cell cycle machinery[22, 23], we analyzed Myosin relocalization timing in *dlg;;pins* double mutants. Compared to wild-type, apical Myosin disappeared later, whereas basal Myosin cleared earlier in *dlg;;pins* mutant neuroblasts. The time difference between apical and basal Myosin clearing decreased significantly (Fig. 1h and Supplementary Fig. 1b, c). Taken together, we conclude that cell cycle and polarity cues regulate the onset and temporal sequence of Myosin relocalization in *Drosophila* neuroblasts.

**Apical and basal Myosin relocalize to the cleavage furrow.** The high temporal resolution live cell imaging results suggest that equatorial Myosin originates from both the apical and basal cortex. To test this hypothesis, we devised photoconversion experiments to investigate the fate of apical and basal Myosin molecules. To this end, we generated transgenic flies, expressing either the regulatory subunit (encoded by *spaghetti squash; sqh*[24]) or the Myosin heavy chain (encoded by *zipper; zip*[25]) tagged with the photoconvertible fluorescent protein mDendra2[26]; both constructs are expressed by endogenous regulatory elements (see methods). We obtained identical results with both lines and will collectively call these fusion proteins Myo::mDendra2 hereafter. We first photoconverted Myo::mDendra2 selectively on the apical cortex shortly before apical Myosin clearing and followed the subsequent relocalization of these photoconverted filaments with live cell imaging in intact fly larval brains or isolated neuroblasts (see methods). This pool of photoconverted Myosin spread almost over the entire cortex and subsequently focused at the cleavage furrow region (Fig. 2a, Supplementary Fig. 2m, Supplementary Movies 1 and 2). Similarly, Myo::mDendra2 filaments that were photoconverted on the basal cortex in early anaphase accumulated at the forming cleavage furrow later in anaphase

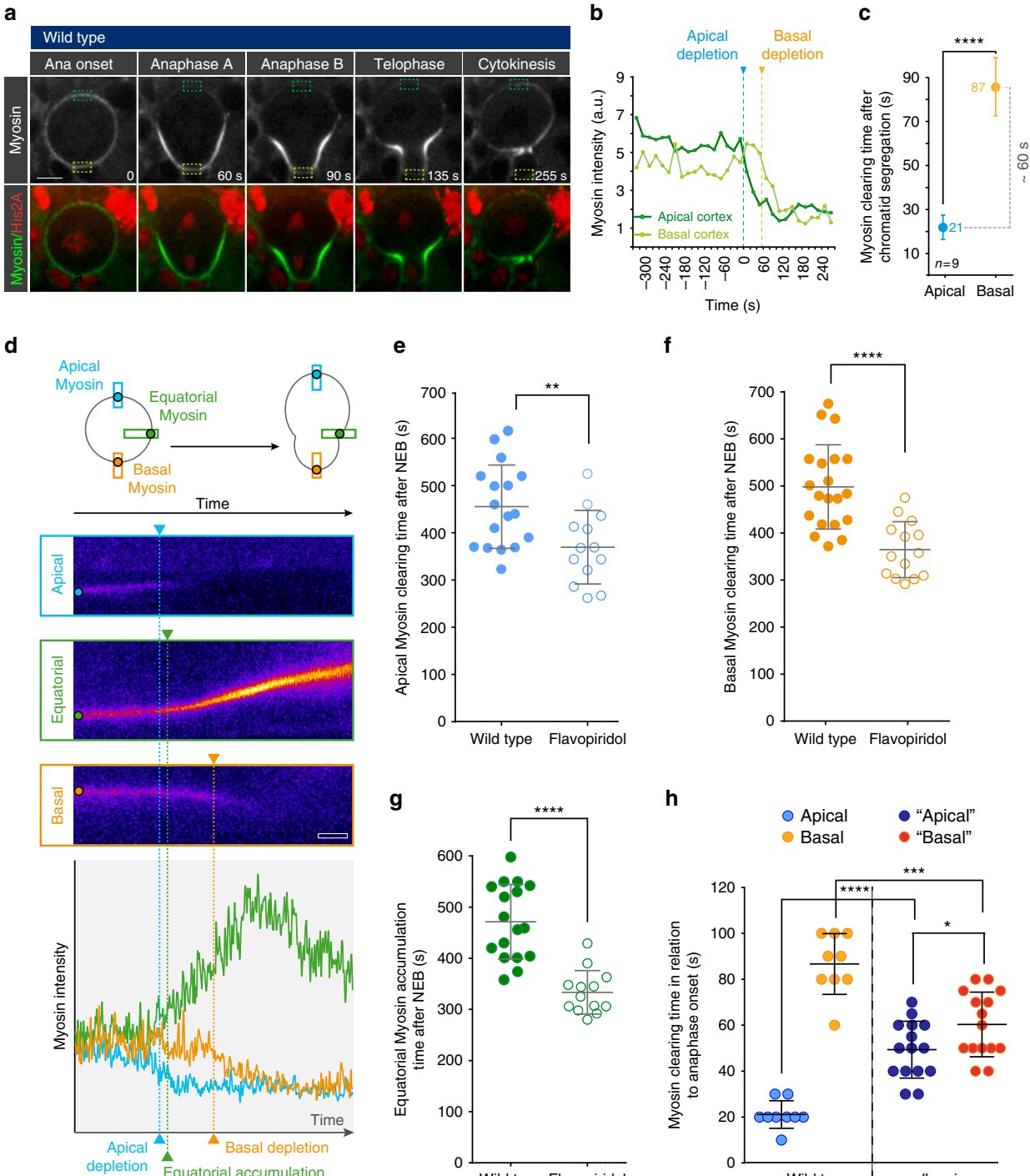

**Fig. 1** Cell cycle and polarity cues regulate Myosin's dynamic relocalization. **a** Representative image sequence showing a wild-type neuroblast expressing Sqh::GFP (Myosin, white; top row, green; bottom row) and the DNA marker His2A::mRFP (red; bottom row). Cortical Myosin intensity was measured at the apical (dark green dashed box) and basal cortex (light green dashed box) throughout mitosis and plotted in **b**. Chromatid segregation starts at "0 s". **c** Mean apical and basal Myosin clearing time and standard deviation in relation to chromatid segregation. The blue and orange numbers represent the mean value. **d** Kymographs showing Myosin intensity at the apical (blue boxes), lateral (green boxes) and basal neuroblast cortex (orange boxes) for one representative wild-type neuroblast. Kymographs were generated from high temporal resolution time-lapse movies (2 s acquisition time). The graph shows Myosin intensity at the apical (blue plot), lateral (green plot) and basal cortex (orange plot). Apical **e** basal **f** Myosin clearing time after nuclear envelope breakdown (NEB) in control and Flavopiridol-treated neuroblasts. **g** Equatorial Myosin accumulation time after NEB. **h** Scatter plot showing the apical and basal Myosin clearing time in relation to anaphase onset, for wild-type and *dlg;;pins* mutant neuroblasts. For this and all subsequent figures: since polarity is lost in *dlg;;pins* mutants, we refer to the cortex clearing slightly earlier or associated with the slightly bigger cell as "apical", whereas the other cortex is referred to as "basal". Center values and error bars represent the mean and standard deviation (s.d), respectively. Asterisks denote statistical significance, derived from unpaired t-tests: *$p \leq 0.05$, **$p \leq 0.01$, ***$p \leq 0.001$, ****$P \leq 0.0001$. Each measured cell (n) is represented with a dot in the scatter plots. For other graphs, the number of measured cells is indicated in the corresponding panels. For each experiment, the data was collected from at least 3 independent experiments. For each independent experiment, at least 5 larvae were dissected. Time: seconds (s). Scale bar: 5 μm. Time scale bar (open white box) in (d): 20 s. n.s. not significant

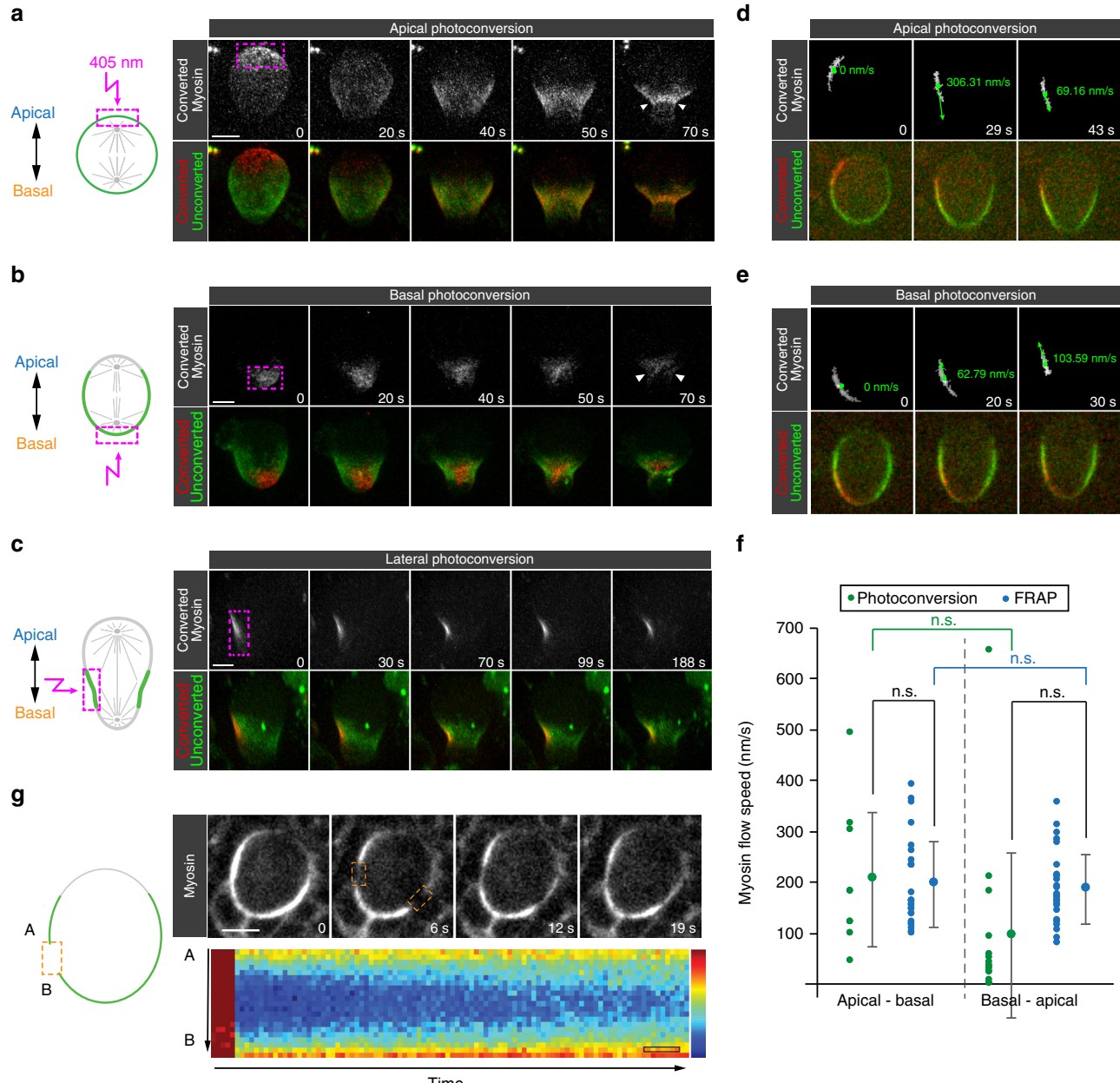

**Fig. 2** Myosin relocalizes to the cleavage furrow through cortical flow. Myosin::Dendra2 was photoconverted on the (**a**) apical, (**b**) basal and (**c**) lateral neuroblast cortex. Top row: maximal projection of photoconverted Myosin. Bottom row: overlay between non-photoconverted (green) and photoconverted Myosin (red), shown in maximal projection. The purple dashed box represents the photoconverted ROI. **d** Representative wild-type neuroblasts showing photoconverted Myosin at the apical (top) or **e** basal cortex (bottom). Top row: maximal projection of photoconverted Myosin. Bottom row: maximal projection showing both non-photoconverted (green) and photoconverted Myosin (red). Myosin flow velocity was determined with custom-made software and plotted in **f**. Center values and error bars represent the mean and standard deviation (s.d), respectively. Asterisks denote statistical significance, derived from unpaired $t$ tests: $*p \leq 0.05$, $**p \leq 0.01$, $***p \leq 0.001$, $****p \leq 0.0001$. Each measured cell (n) is represented with a dot in the scatter plots. For each experiment, the data were collected from at least 3 independent experiments. For each independent experiment, at least 5 larvae were dissected. **g** Representative wild-type neuroblast expressing Sqh::GFP (Myosin; white). The GFP signal was bleached in two regions of interests (ROIs; orange dashed boxes) and kymographs (shown below the image sequence) were used to measure the recovering fluorescence. The shown kymograph was derived from the left lateral side as shown in the cartoon. Highest fluorescence intensity is shown in red, lowest in blue. Time: seconds (s). Scale bar: 5 μm. Time scale bar (open black box) in (g): 1 s. n.s. not significant

(Fig. 2b, Supplementary Fig. 2m and Supplementary Movie 3). Thus, consistent with the high temporal live cell imaging results, both pre-anaphase apical and early anaphase basal Myosin molecules contribute to the forming contractile ring.

**Myosin filaments flow towards the cleavage furrow region.** Apical and basal cortical Myosin filaments could locally disassemble and redistribute through the cytoplasm, thereby reaching the cleavage furrow. Alternatively, the contractile properties of the actomyosin cytoskeleton could induce the onset of first an apical—basal (basal directed) and subsequently a basal—apical (apical directed) cortical flow[27]. To distinguish between these scenarios, we converted Myo::mDendra2 at the lateral cortex shortly after apical clearing. If cortical Myosin

filaments would contribute to the cleavage furrow through cytoplasmic relocalization, we anticipated that Myosin would label the furrow symmetrically. Alternatively, laterally photo-converted Myosin filaments should predominantly stay at the lateral neuroblast cortex if cortical flow is a major mechanism (Supplementary Fig. 2a, b). In all cells (100%; $n = 49$), we observed that photoconverted Myosin remains asymmetrically localized after photoconverting in early anaphase neuroblasts, labeling predominantly one side of the neuroblast cortex during anaphase and early telophase. Furthermore, apically

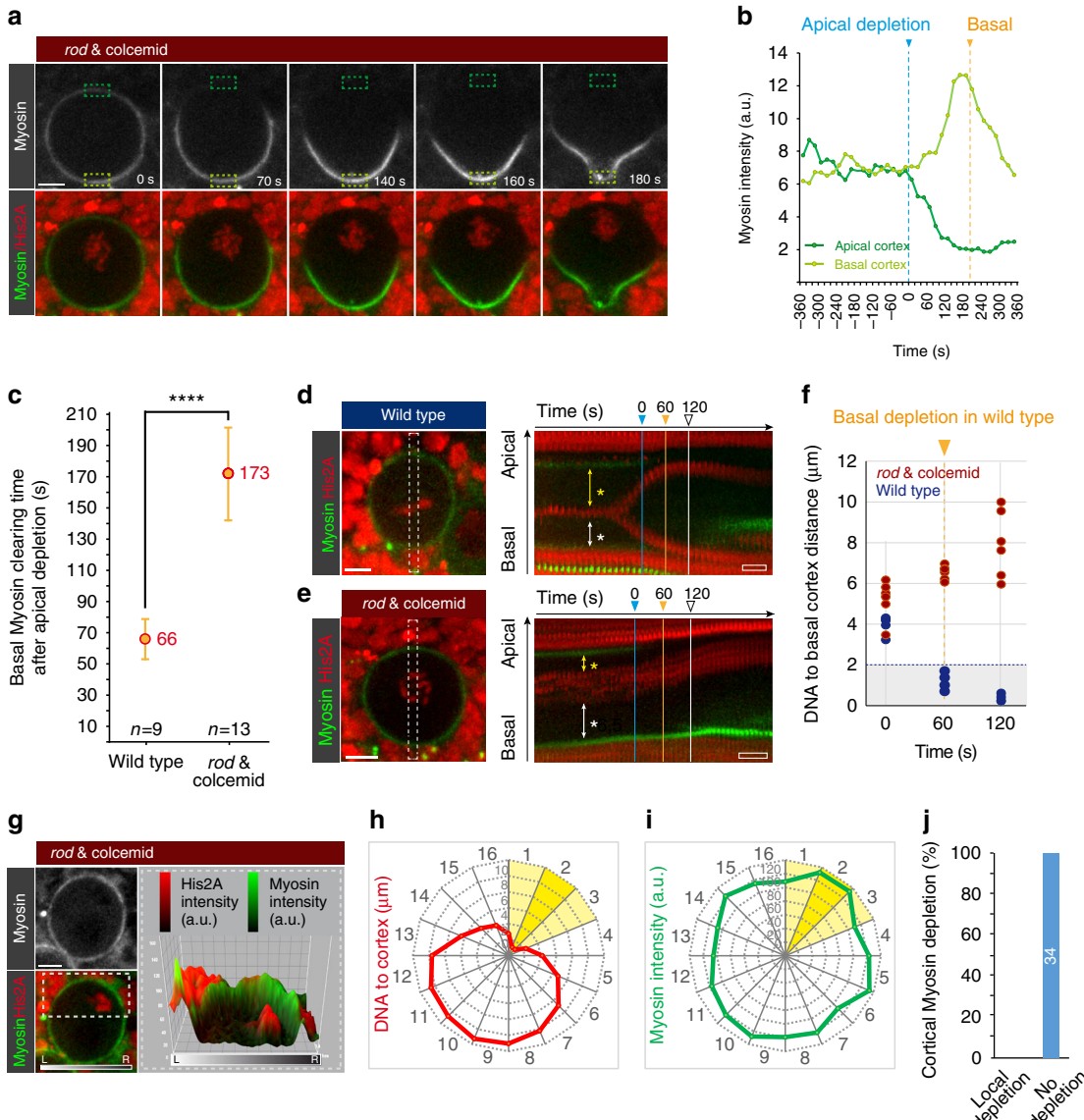

**Fig. 3** Chromatid-derived cues are insufficient to induce Myosin relocalization in fly neuroblasts. **a** Representative image sequence showing a *rod* mutant neuroblast expressing Sqh::GFP (Myosin, top row; white, bottom row; green) and His2A::mRFP (DNA, bottom row; red) exposed to colcemid. Cortical Myosin intensity was measured at the apical and basal cortex, respectively (dark and light green dashed boxes) and plotted in (**b**). **c** Graph showing the mean and standard deviation of basal Myosin clearing in relation to apical depletion (timepoint "0") for wild-type and *rod* mutant, colcemid-treated neuroblasts. Kymographs obtained from wild-type (**d**) and *rod* mutant neuroblasts exposed to colcemid (**e**) showing Myosin (green) and DNA (red). The white dashed line indicates the region represented in the kymograph. Kymographs were used to measure the distance between the chromosomes and the apical (yellow arrows) or basal cortex (white arrows). **f** Scatter plot representing the distance between chromosomes and the basal cortex at the onset of apical Myosin depletion (0 s), at the onset of basal Myosin clearing (60 s) and once basal Myosin depletion is completed (120 s). **g** Representative *rod* mutant, colcemid-treated neuroblast expressing Sqh::GFP (white; top panel, green; merge) and His2A::mRFP (red). Myosin and His2A intensity are shown in the 3D graph, corresponding to the ROI represented by the white dashed box. Radar graphs show the distance of the neuroblast's chromatid to the cortex (**h**) and Myosin intensity (**i**) for each of the 16 represented sectors. Yellow sectors represent areas with high DNA-cortex proximity. **j** Bar graph showing the percentage of *rod* mutant, colcemid treated neuroblasts displaying cortical Myosin depletion when the proximity of the DNA with the cortex is <2 μm. Center values and error bars represent the mean and standard deviation (s.d), respectively. Asterisks denote statistical significance, derived from unpaired *t* tests: *$p \leq 0.05$, **$p \leq 0.01$, ***$p \leq 0.001$, ****$p \leq 0.0001$. Each measured cell (n) is represented with a dot in the scatter plots. For other graphs, the number of measured cells is indicated in the corresponding panels. For each experiment, the data were collected from at least 3 independent experiments. For each independent experiment, at least 5 larvae were dissected. Time: seconds (s). Scale bar: 5 μm. Time scale bars (open white box): 57 s in (**d**) and 60 s in (**e**), respectively. n.s. not significant

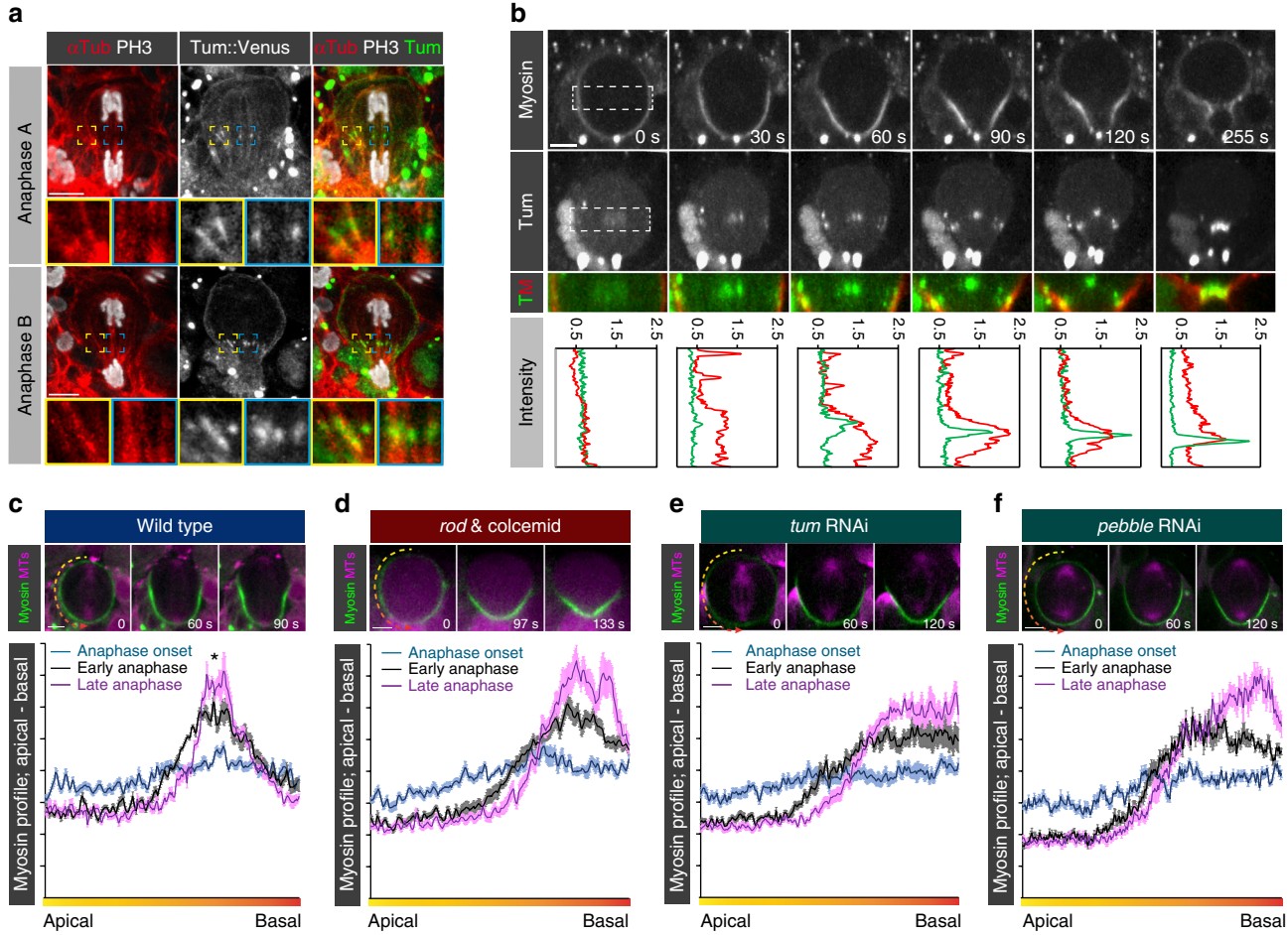

**Fig. 4** The centralspindlin complex induces basolateral Myosin enrichment, necessary for basal Myosin relocalization. **a** Representative wild-type neuroblasts expressing Tum::Venus (white; middle row, green; overlay), and stained for alpha-Tubulin (αTub; red) and phospho-Histone3 (PH3; white in first and third row). Higher magnification images correspond to regions highlighted with yellow and blue squares. **b** Representative image sequence of a wild-type neuroblast expressing Myosin (white; top row, red; overlay in third row) and Tumbleweed (white; second row, green; overlay in third row). Higher magnification pictures were taken from the regions highlighted with white dashed boxes and shown as a merge (third row). Myosin (red) and Tumbleweed (green) intensity plots, obtained from the apical to the basal cortex, are shown for each time point. Representative image sequences of third instar neuroblasts expressing Sqh::GFP (Myosin; green) and Cherry::Jupiter (MTs; purple) for (**c**) wild type, (**d**) colcemid-treated *rod* mutant, (**e**) Tum or (**f**) Pebble depleted neuroblasts. Myosin intensity was measured from the apical to the basal neuroblast cortex at three different time points. For all conditions, the mean intensity, derived from 5 cells and standard deviation is plotted. The data were collected from at least 3 independent experiments. The star corresponds to the lateral Myosin enrichment detected in early anaphase in wild-type neuroblasts. Scale bars: 5 μm

and laterally photoconverted Myo::mDendra2 flowed to the furrow region and the photoconverted patch became more confined at the onset of furrow ingression (Fig. 2c, d and Supplementary Movie 4). We developed software to quantify Myosin flow velocity from these photoconversion data sets (see methods and Supplementary Fig. 2c). Velocity measurements were only performed until cell deformation set in to exclude an overestimation of Myosin flow speed. These measurements did not reveal a statistical significant difference between the basally and apically directed Myosin flow, albeit some variability was detected (Apical—Basal: 206.8+/− 131 nm/s, $n = 9$. Basal—Apical: 96.4+/− 160 nm/s, $n = 15$; Fig. 2d–f). We also performed FRAP experiments in metaphase and early anaphase neuroblasts, measuring the recovery of Sqh::GFP in the bleached region. In contrast to metaphase, we found that Myosin filled in the bleached region from both the apical and the basal edge, supporting the photoconversion data and confirming the existence of a basal- and apical-directed Myosin flow (Fig. 2g, f and Supplementary Fig. 2d, e). FRAP experiments on neuroblasts expressing membrane tethered mCherry (mCherry::CAAX) showed much faster

recovery dynamics than simultaneously bleached Sqh::GFP (Supplementary Fig. 2f–i), which was expected for a membrane-associated protein not restricted in its lateral movement by the actin cytoskeleton[28, 29]. Myosin flow velocity quantifications from kymographs revealed values comparable to the photoconversion data sets (Apical—Basal: 196.7+/− 84.3 nm/s, $n = 26$; Basal—Apical: 187.3+/− 68.9 nm/s, $n = 26$; Fig. 2f). Overall, the two converging Myosin flows showed no statistically significant velocity difference.

The observed flows are not a consequence of cell shape changes since photoconverted Gap43::mEos[30]—a membrane marker—from the apical, basal or lateral neuroblast cortex distributed over the entire neuroblast membrane. Unlike Myosin, photoconverted Gap43::mEos did not specifically enrich at the cleavage furrow (Supplementary Fig. 2j–m, Supplementary Movie 5 and 6). Although we cannot directly pinpoint the spatial origin, these results strongly suggest that Myosin's dynamic relocalization is due to two cortical flows: an apical—basal flow and a basal—apical flow. Furthermore, spatiotemporally controlled down- or upregulation of Myosin activity could precede both

the apically and basally directed flows. Taken together, we conclude that cortical flow is a major mechanism to relocalize Myosin filaments from both poles to the cleavage furrow in mitotic fly neuroblasts but do not exclude the contribution of cytoplasmic Myosin molecules to the cleavage furrow during anaphase (see also below).

**Myosin clearing is independent of chromatin derived cues.** Next, we investigated the molecular mechanisms underlying spatiotemporally controlled Myosin flows. The contribution of cell polarity to Myosin relocalization, as described above, is reported elsewhere[31]. Here we focus on how the mitotic spindle induces Myosin relocalization.

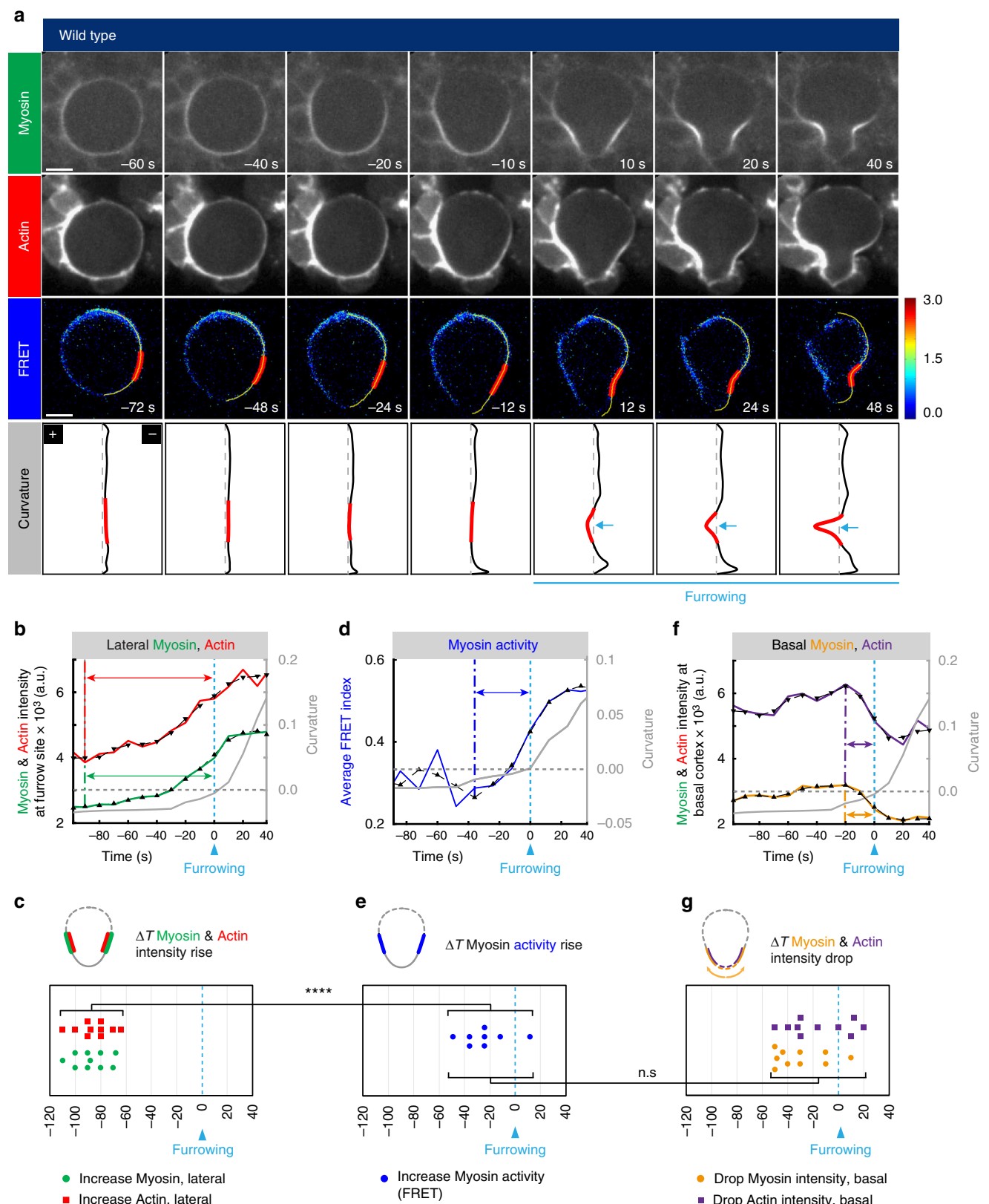

Neuroblasts lacking mitotic spindles (colcemid-treatment) and the spindle-assembly checkpoint component Rod[32], displayed a strong delay in basal Myosin clearing[12, 18, 20] (and Fig. 3a–c). In addition, the neuroblast's chromatin failed to reach the basal cortex (Fig. 3a). Since chromatin-derived cues have recently been implicated in cortical remodeling[33, 34], we investigated its role during asymmetric cell division in more detail. In wild-type neuroblasts, chromatids approached the basal cortex more than the apical cortex (Fig. 3d and Supplementary Fig. 3a). In colcemid-treated *rod* mutant neuroblasts, the chromosomes stayed relatively close to the apical cortex, progressively moving away from the basal cortex during anaphase (Fig. 3e, f and Supplementary Fig. 3b).

On the basis of these data, we tested a potential connection between DNA-derived cues and basal Myosin relocalization. We first tested whether the small GTPase Ran, the phosphatase Pp1-87B and its regulatory subunit Sds22—all of which were previously implicated in chromatid-associated cortex remodeling[33–35]—are required for apical and basal Myosin clearing. Ran accumulated around the chromatin in metaphase as reported for other cell types[36–38] but subsequently enriched on the sister chromatids, segregating into the neuroblast (Supplementary Fig. 3c). Pp1-87B was associated with neuroblast chromatin in interphase and late telophase but was widely distributed throughout the neuroblast's cytoplasm during anaphase when Myosin relocalization starts (Supplementary Fig. 3f). Thus, neither Ran's nor Pp1-87B's localization correlated with the sequence of basal Myosin clearing. Knocking-down *Ran, Pp1-87B* or *sds22* with inducible RNAi, expression of the dominant-negative Ran[T24N] [39] or using mutant alleles to remove *sds22* and *Pp1-87B*, respectively did neither compromise apical nor basal Myosin clearing (Supplementary Fig. 3d, e, g–j). However, we noticed that sibling cell size asymmetry was perturbed after knocking-down Sds22 (see below).

Finally, to exclude the involvement of another chromosome-derived signal, we treated neuroblasts with colcemid or induced local membrane and cell cortex lesions in metaphase neuroblasts to artificially push chromatin close to the cell cortex. Nevertheless, we did not observe local Myosin depletion in both experiments (100%; $n = 34$; Fig. 3g–j and Supplementary Fig. 3k–m).

Taken together, we conclude that neither Ran, Pp1-87B, Sds22 nor any other chromatin-derived cues are necessary to induce local Myosin depletion. Furthermore, chromatin-derived cues are not sufficient to clear cortical Myosin in mitotic neuroblasts. Thus, the lack of basal Myosin clearing in colcemid treated neuroblasts is directly related to a lack of spindle-dependent cues.

**The central spindle pathway induces the basal Myosin flow.** We next asked how the mitotic spindle could induce an apically

directed Myosin flow. The equatorial stimulation model proposes that microtubules contacting the equatorial cortex (of central spindle or astral origin) lead to Myosin activation at the cell equator through centralspindlin-dependent activation of RhoA[5]. Due to the intrinsic contractile properties of Myosin, such an increase in activated Myosin generates a cortical flow towards the highest Myosin density[40]. To test whether this model could explain the basal—apical Myosin flow in neuroblasts, we first analyzed the localization of the centralspindlin complex using Tum::Venus[41]. From anaphase onset onwards, Tum was detected on bundled microtubules in the cell center, but also decorated microtubules contacting the lateral cortex (Fig. 4a, b and Supplementary Fig. 4a). Importantly, Tum preceded Myosin's focused enrichment on the lateral cortex (Fig. 4b; compare 60 s vs. 90 s time point). Subsequently, both Tum and Myosin enrichment shifted closer to the basal cortex during anaphase (Fig. 4b; Timepoints 60 s–255 s). Tum localization and subsequent Myosin enrichment agrees with the equatorial stimulation model. To confirm it, we (1) depleted the mitotic spindle completely (rod[H4.8] mutants treated with colcemid), (2) removed the centralspindlin components Tum and Pav, (3) knocked-down the CPC component AurB, acting upstream of the centralspindlin complex[42] and (4) removed the RhoGEF Pebble (Ect2 in mammals). Apical Myosin relocalization is not affected under these conditions[20] but lateral enrichment was abolished. Furthermore, Myosin enriched on the basal cortex and cleared with a significant delay (Fig. 4c–f and Supplementary Fig. 4b–d). We further tested whether premature activation of the central spindle pathway can induce an apically directed Myosin flow earlier. Partial Cdk1 inactivation has been shown to be sufficient to promote central spindle formation and to activate the spindle-dependent equatorial Myosin activation pathway[21, 43]. Indeed, in relation to apical Myosin clearing, neuroblasts exposed to Flavopiridol accumulated Myosin earlier at the prospective cleavage furrow and prematurely induced basal Myosin clearing. Premature basal Myosin clearing reduced the time window between apical and basal Myosin clearing, similar to *dlg;;pins* mutant neuroblasts (Fig. 1h, Supplementary Fig. 1b, c and Supplementary Fig. 4e–h). Taken together, we conclude that the central spindle pathway is necessary for basal Myosin relocalization.

**Actomyosin enriches laterally prior to furrowing.** Since cortical Myosin is bound to filamentous Actin (F-Actin), locally activated Myosin should induce an F-Actin flow. Thus, we reasoned that F-Actin should relocalize with similar dynamics than Myosin and therefore also clear on the basal cortex after Myosin accumulated at the prospective furrow region. To test this hypothesis, we analyzed the dynamics of Lifeact—a probe for F-Actin[44]—in wild-type neuroblasts. Similar to Myosin, F-Actin accumulation in the prospective furrow region started about 120—60 s prior to

**Fig. 5** Lateral Actomyosin enrichment precedes basal actomyosin clearing. **a** Representative image sequence of a wild-type neuroblast expressing Sqh::mCherry (Myosin), LifeAct::GFP (Actin) and the Myosin activity FRET sensor. Curvature is determined for each pixel along the yellow line (4th row). Ingression occurs when curvature changes from "−" to "+". The region corresponding to the future cleavage furrow is indicated in red and plotted in the graphs below (grey line). Blue arrows emphasize furrowing. **b** Plot of a representative neuroblast showing Myosin (green line) and Actin (red line) intensity changes in the prospective furrow region, in relation to curvature changes. For this and subsequent plots, curves were smoothened (dashed black line with black triangles) and plotted in the same graph. The time difference between an increase in Myosin and Actin intensity in the prospective furrow region and furrowing (dashed lines and green and red arrows, respectively), was extracted from the smoothened curves and plotted in (**c**). **d** Plot showing changes in FRET ratios in the prospective furrow region in relation to furrowing. The time difference between the onset of FRET ratio increases and furrowing (dashed blue lines and blue arrows, respectively) are shown in **e**. **f** Actin (purple line) and Myosin (orange line) intensity changes were plotted for the basal cortex. The time difference between actomyosin intensity drops and furrowing (dashed lines and purple and orange arrows, respectively) were plotted in **g**. Asterisks denote statistical significance, derived from unpaired *t* tests: *$p \leq 0.05$, **$p \leq 0.01$, ***$p \leq 0.001$, ****$p \leq 0.0001$. Each measured cell (n) is represented with a dot in the scatter plots. For each experiment, the data were collected from at least 3 independent experiments. For each independent experiment, at least 5 larvae were dissected. Time: seconds (s). Scale bars: 5 μm. n.s. not significant

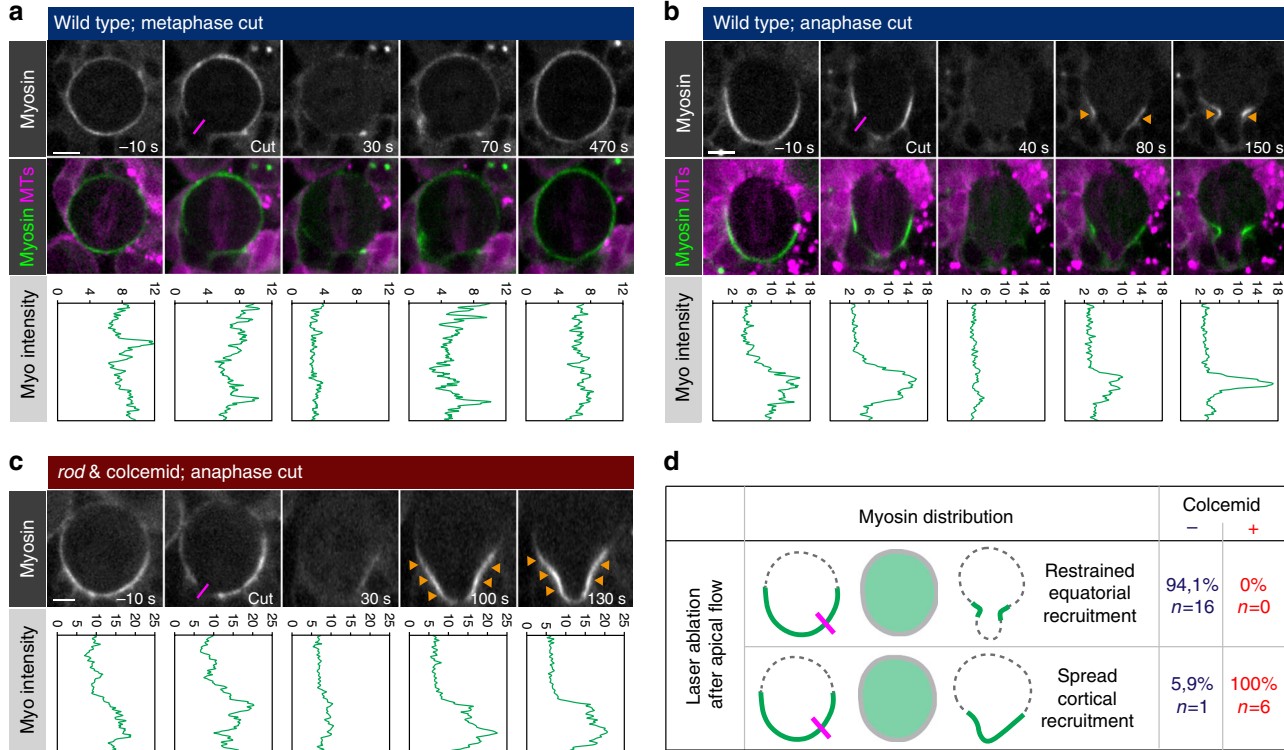

**Fig. 6** The anaphase spindle is necessary for focusing Myosin localization at the cleavage furrow. **a** Laser-induced cortical cutting experiments performed on third instar wild-type neuroblasts expressing Sqh::GFP (Myosin, white; upper panel, green; lower panel) and Cherry::Jupiter (MTs; magenta) in metaphase, or **b** anaphase. For this and subsequent panels, the cortical cutting site is highlighted with purple lines. Orange arrowheads highlight Myosin recruitment. Myosin intensity profiles, measured from the apical to the basal cortex, are shown underneath the corresponding image. **c** Laser-induced cortical cutting experiments performed on third instar *rod* mutant neuroblasts exposed to colcemid in anaphase. Only Sqh::GFP (Myosin; white) is shown. **d** Experimental summary. Time is shown as seconds before and after cortical cutting. Scale bars: 5 μm

furrow ingression (Fig. 5a–c). Increase in Myosin and F-Actin should result in an increase in active tension. To test this idea, we constructed a Myosin activity sensor by adding two Vinculin domains that bind filamentous Actin (F-Actin) and a FRET module, separated by a flexible spider silk protein (31, 45. See also methods). If Actin filaments are pulled together due to Myosin motor activity the sensor will respond with high FRET signals (Supplementary Fig. 5a, b). In most cases, this sensor showed an increase in FRET ratios between 10–50 s prior to furrow ingression in wild-type neuroblasts (Fig. 5a, d, e). Neuroblasts, treated with the Rho kinase inhibitor Y27632 showed a significant reduction in FRET ratios in the cleavage furrow region although ingression was initiated (Supplementary Fig. 5c–f), suggesting that the sensor accurately, albeit indirectly, monitors Myosin activity.

Per equatorial stimulation model, increase of activated Myosin should precede or coincide with basal actomyosin clearing. We detected a concomitant intensity drop for both Myosin and F-Actin on the basal neuroblast cortex, concurring with the detection of activated Myosin in the prospective furrow region but shortly before furrowing initiated (Fig. 5d–g). Taken together, these measurements show that Myosin and F-Actin first accumulated at the lateral cortex, followed by an increase in FRET signal—an indirect readout for active tension—in the prospective furrow region and a concomitant decrease of Myosin and F-Actin on the basal cortex. These data suggest that the increase in activated Myosin at the lateral cortex could be the motor for basal Myosin clearing.

**The anaphase spindle is focusing Myosin at the lateral cortex.** Our photoconversion experiments showed that directed Myosin flows contribute to lateral Myosin enrichment but we wanted to

know whether Myosin can also be recruited to the equatorial cortex from the cytoplasm. Cytoplasmic recruitment could contribute towards the lateral enrichment of Myosin as shown above. To this end, we used a pulsed UV laser to induce local lesions in the neuroblast cortex (see methods). Cutting the cortex caused cortical Myosin to fall into the cytoplasm within ~40 s, thereby creating a neuroblast cortex that is essentially devoid of cortical Myosin. Under these conditions, the spatiotemporal relocalization of Myosin can be followed. Indeed, 1–2 minutes after the metaphase cortex was cut, cytoplasmic Myosin returned to the neuroblast cortex with a uniform cortical distribution (Fig. 6a and Supplementary Fig. 6a, b). However, if the cortex was cut in early anaphase (after apical Myosin depletion), Myosin also returned to the cortex but did not spread uniformly anymore; Myosin was localized in a confined band, coinciding with the ingressing cleavage furrow (Fig. 6b, d and Supplementary Movie 7). Surprisingly, in neuroblasts devoid of mitotic spindles (colcemid-treated *rod* mutants) Myosin was still able to return to the cortex after cutting. However, furrow confinement was lost and Myosin spread out on the basal cortex (Fig. 6c, d and Supplementary Movie 8). We conclude that (1) cytoplasmic Myosin can be recruited to the equatorial cortex independently of the mitotic spindle but (2) spindle-dependent cues are required to focus Myosin to the cleavage furrow region.

**Regulated Myosin flows contribute to physical asymmetry.** Since unequal cortical expansion is dependent on Myosin localization[18] the differential Myosin flow onset described here would provide an intuitive model for basal cleavage furrow positioning and thus physical asymmetry. To test this hypothesis,

we identified mutant conditions, altering Myosin flow onset on both poles and tested whether these manipulations altered physical asymmetry (Fig. 7a). In wild-type neuroblasts, the temporal difference between the two flows is ~ 60 s (Mean: 65.56 s;+/−

13.33; $n = 9$) although significant variations between individual cells exist (Fig. 7b, c). *rod* mutant neuroblasts treated with colcemid showed a strong delay in basal Myosin clearing, increasing the time between apical and basal Myosin flow onset to almost 3

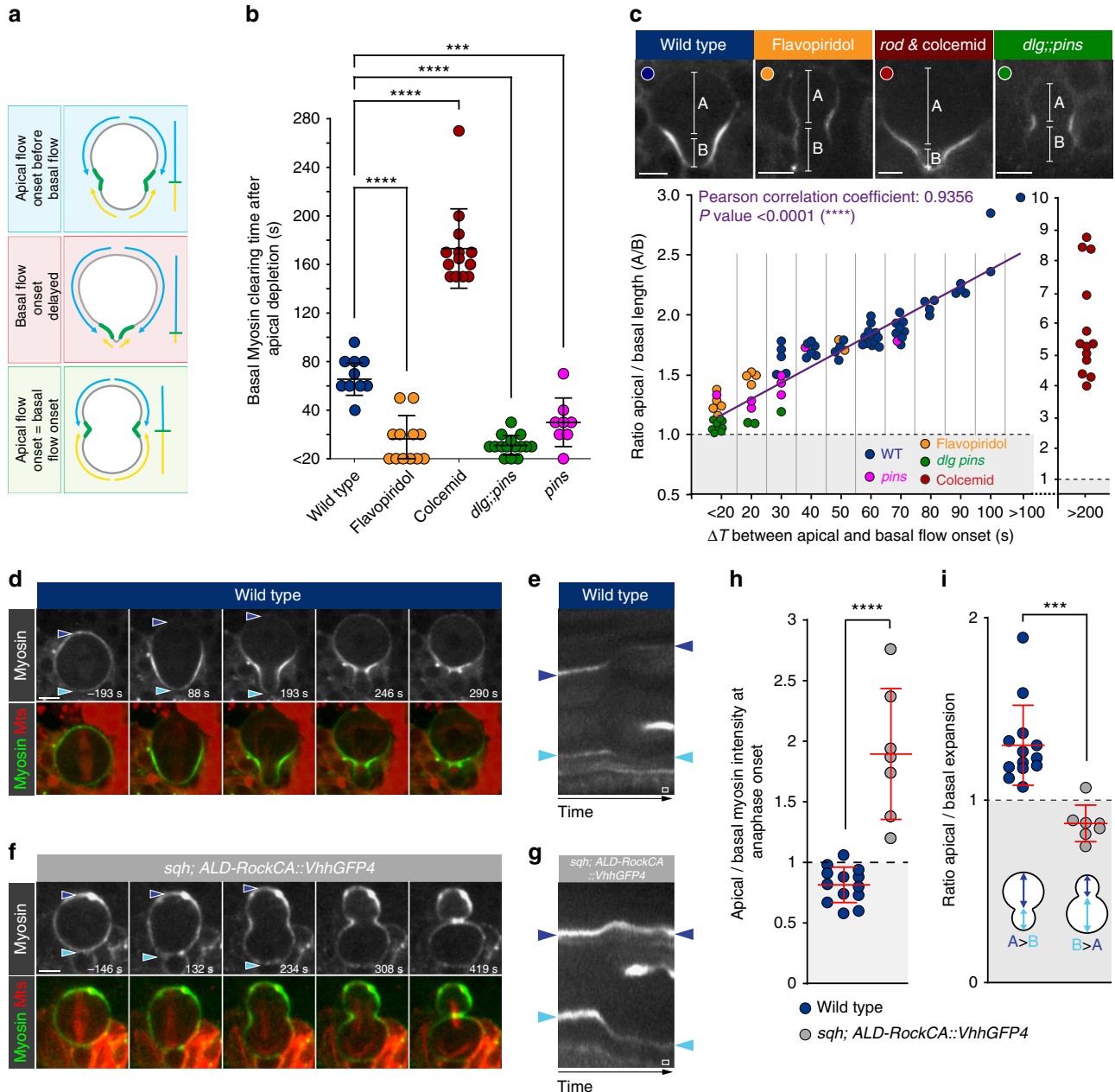

**Fig. 7** Myosin flows are instrumental in establishing sibling cell size asymmetry. **a** Schematic illustration of the model to be tested: if cleavage furrow positioning and thus physical asymmetry depends on Myosin flows, then altering Myosin flow onset on the apical and/or basal cortex will misposition the cleavage furrow. **b** Scatter plot showing basal Myosin clearing time in relation to apical Myosin relocalization in wild-type, neuroblasts exposed to Flavopiridol or colcemid, *dlg;;pins* and *pins* mutants. **c** Scatter plot showing the correlation between Myosin clearing on the apical and basal cortex and the asymmetry of the division in anaphase. A Pearson coefficient of "1" indicates a perfect correlation between Myosin clearing and furrow positioning. **d** Representative image sequence and (**e**) kymograph showing a wild-type neuroblast (endogenous untagged Sqh still present) expressing Sqh::GFP (Myosin, white; top row, green; bottom row) and Cherry::Jupiter (MTs; red; bottom row). **f** Representative image sequence and **g** kymograph of a *sqh* mutant neuroblast, coexpressing Sqh::GFP and ALD-Rock[CA]::VhhGFP4, a fusion between the single-chain GFP antibody (VhhGFP4), Inscuteable's apical localization domain (ALD) and Rho kinase's kinase domain. **h** Kymographs were used to calculate apical/basal Myosin intensity ratios and (**i**) apical and basal cortical expansions. The resulting ratios are represented as a scatter plot. Center values and error bars represent the mean and standard deviation (s. d.), respectively. Asterisks denote statistical significance, derived from unpaired t-tests: *$p \leq 0.05$, **$p \leq 0.01$, ***$p \leq 0.001$, ****$p \leq 0.0001$. Each measured cell (n) is represented with a dot in the scatter plots. For each experiment, the data were collected from at least 3 independent experiments. For each independent experiment, at least 5 larvae were dissected. Time: seconds (s). Scale bar: 5 µm. Time scale bar (open white box) in (**e**, **g**): 58 s. n.s. not significant

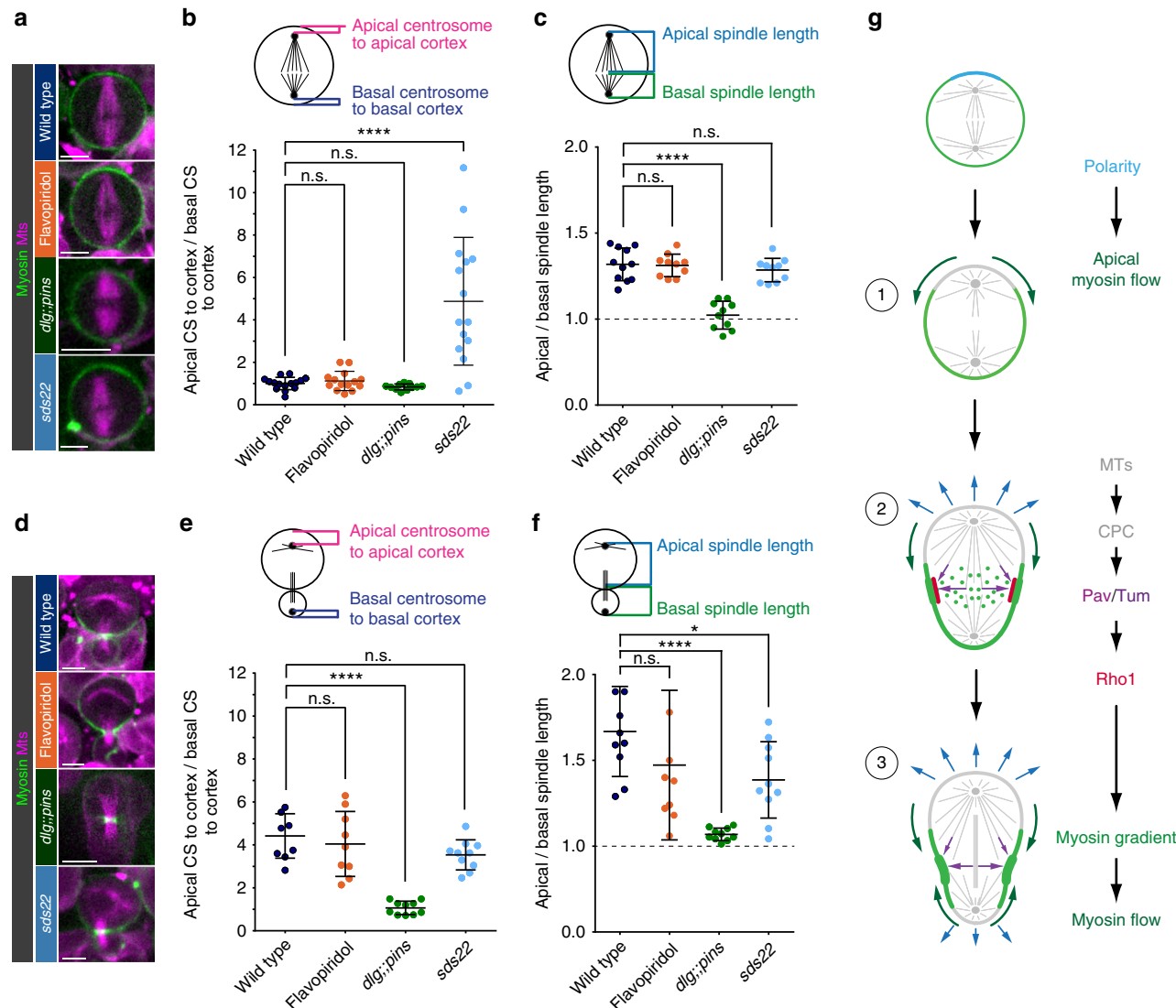

**Fig. 8** Spindle asymmetry and positioning contribute to cleavage furrow placement and physical asymmetry. Representative snapshots for wild type, Flavopiridol-treated, *dlg;;pins* and *sds22* mutant (**a**) metaphase and (**d**) telophase neuroblasts. Myosin is shown in green and MTs in magenta. **b**, **e** Scatter plots showing spindle position and (**c**, **f**) spindle asymmetry for each genotype at metaphase and telophase, respectively. **g** Model. In *Drosophila* neuroblasts, polarity cues are used to initiate an apical—basal Myosin flow. Apical clearing of Myosin allows apical cortical expansion. Subsequently, Myosin flows from the basal towards the apical neuroblast cortex induced through a lateral Myosin gradient, which is established through anaphase spindle cues. The chromosomal passenger and centralspindlin complexes are required to enrich for Myosin on the lateral cortex, initiating basal Myosin clearing. Spindle positioning and geometry are key factors to localize the site of cleavage furrow formation, confining Myosin enrichment to the lateral cortex. See text for details. Center values and error bars represent the mean and standard deviation (s.d), respectively. Asterisks denote statistical significance, derived from unpaired *t* tests: *$p \le 0.05$, **$p \le 0.01$, ***$p \le 0.001$, ****$P \le 0.0001$. Each measured cell (n) is represented with a dot in the scatter plots. For each experiment, the data were collected from at least 3 independent experiments. For each independent experiment, at least 5 larvae were dissected. Scale bars: 5 µm. n.s. not significant

minutes (Mean: 173.1 s;+/− 32.69; *n* = 13; Fig. 7b). Neuroblasts depleted for intrinsic polarity such as *pins* single or, as shown above, *dlg;;pins* double mutants displayed a reduced delay between apical and basal Myosin clearing (Mean: 11 s;+/− 8.062; *n* = 15; Fig. 1h, Fig. 7b and Supplementary Fig. 1b, c). Similarly, the delay between apical and basal Myosin flow onset was minimal in Flavopiridol treated neuroblasts (Mean: 16.36 s;+/− 19.12; *n* = 11; Fig. 7b and Supplementary Fig. 4e, f), mimicking *dlg;;pins* mutant neuroblasts.

Myosin relocalization times also correlated well with cortical expansion. Wild-type neuroblasts showed more apical than basal cortical expansion. In colcemid-treated neuroblasts only basal

growth was reduced, also due to a retraction of the cortex. *dlg;; pins* double mutants and Flavopiridol-treated cells showed comparable cortical expansion ([18]; Supplementary Fig. 8a, b).

Next, we correlated Myosin relocalization timing with the establishment of physical asymmetry. To this end, we measured the distance from the cleavage furrow to the apical and basal cortex, respectively to determine an asymmetry ratio and plotted it against the individual clearing time delay. The measured clearing times showed a good correlation with the resulting asymmetry index; the larger the difference between apical and basal clearing, the bigger the asymmetry index. For instance, Myosin clearing times varied between individual wild-type cells

and the resulting sibling cell size asymmetry ratio increased from 1.5–3 with increasing clearing time differences. *dlg;;pins* double mutants showed an asymmetry index close to 1. Colcemid treated *rod* mutant neuroblasts delayed basal Myosin clearing considerably, resulting in an extreme asymmetry ratio. The calculated correlation (Pearson) coefficient is close to 1, suggesting that spatiotemporally regulated Myosin relocalization is a major contributor for physical asymmetry (Fig. 7c).

To further test how Myosin relocalization dynamics influence sibling cell size asymmetry, we modified the recently published anti-GFP nanobody (called VhhGFP4[46] and Caussinus et al., *in preparation*), by functionalizing it with the constitutively active kinase domain from *Drosophila* Rho kinase[45]. We reasoned that biasing Myosin activity to the apical cortex should invert sibling cell size asymmetry, creating a small neuroblast and a large GMC. To this end, we tethered this Rock[CA]::VhhGFP4 with Inscuteable's apical localization domain (ALD;[47], see also methods) and expressed ALD-Rock[CA]::VhhGFP4 in neuroblasts depleted for endogenous *sqh* (Myosin's regulatory subunit). Thus, the entire pool of Sqh is tagged with GFP and becomes susceptible to VhhGFP4 binding. In contrast to wild-type neuroblasts, showing stereotypic apical Myosin clearing and subsequent apical expansion, expressing ALD-Rock[CA]::VhhGFP4 together with Sqh::GFP in *sqh* mutant neuroblasts resulted in a failure to clear apical Myosin and a predominant expansion of the basal side. This inversion of physical asymmetry was predominantly observed in older *sqh* mutant larvae expressing ALD-Rock[CA]::VhhGFP4 together with Sqh::GFP, presumably due to maternal contribution. However, old wild-type or *sqh* mutant larvae expressing Sqh::GFP never showed this physical inversion (Fig. 7d–i, Supplementary Fig. 7a, b, and Supplementary Movie 9). Tethering Rock[CA] (ALD-Rock[CA]:mCherry) to the apical neuroblast cortex without the VhhGFP4 domain was not sufficient to increase apical Myosin phosphorylation (Supplementary Fig. 7c). Expressing ALD-Rock[CA]::VhhGFP4 in either wild-type or *sqh* mutant neuroblasts did not perturb neuroblast intrinsic polarity but showed enrichment of phosphorylated Myosin on the apical neuroblast cortex (Supplementary Fig. 7c, d). ALD-Rock[CA]:: VhhGFP4 could retain Myosin on the apical neuroblast cortex by either trapping Myosin apically, maintain or increase the activity of apical Myosin, or preventing local Myosin clearing and apical expansion due to a combination of both. Taken together, these data strongly suggest that by perturbing Myosin clearing on the apical cortex during anaphase, unequal cortical expansion and thus the establishment of correct physical asymmetry is compromised.

**Spindle asymmetry and positioning refine furrow placement**. We noticed that Flavopiridol-treated and *pins* single mutant neuroblasts had comparable Myosin clearing times to *dlg;;pins*. Nevertheless, in contrast to *dlg;;pins*, the former two conditions only partially reduced physical asymmetry (Fig. 7c and Supplementary Fig. 8d), suggesting that in addition to spatiotemporally regulated Myosin flow onset additional factors contribute to final furrow positioning and sibling cell size asymmetry. A prime suspect is spindle geometry since previous reports correlated spindle asymmetry and positioning with sibling cell size asymmetry[48–50]. We set out to analyze both spindle positioning and spindle asymmetry by measuring the distance of the centrosomes to the cortex (positioning) and the length of the apical and basal spindle half at metaphase and telophase (asymmetry), respectively. We found that compared to wild-type, Flavopiridol-treated and *dlg;;pins* mutant neuroblasts show normally positioned spindles in metaphase. However, metaphase neuroblasts lacking Sds22 displayed spindles that were shifted significantly towards the basal cortex (Fig. 8a, b and Supplementary Fig. 7e).

Interestingly, metaphase spindle asymmetry was normal in all conditions with the exception of *dlg;;pins* mutant neuroblasts, which displayed symmetric spindles (Fig. 8a, c). Although we could not measure spindle asymmetry and positioning during anaphase, we found that in telophase, spindles of wild-type, *sds22* mutants and neuroblasts treated with Flavopiridol, were displaced towards the basal cortex. Only *dlg;;pins* mutants contained centered telophase spindles (Fig. 8d, e). However, telophase asymmetry was mostly affected in *dlg;;pins* and—although overall not significant—also compromised in some Flavopiridol-treated or *sds22* deficient neuroblasts (Fig. 8d, f). Measuring the timing between Myosin clearing at the apical and basal cortex in *sds22* deficient neuroblasts did not show a correlation between Myosin relocalization dynamics and sibling cell size asymmetry (Supplementary Fig. 7f). Spindle geometry correlated well with the shift in furrow positioning (Supplementary Fig. 8c). We conclude that in addition to temporally regulated Myosin flow onset, spindle positioning and spindle asymmetry are important contributors to cleavage furrow positioning and final sibling cell size asymmetry.

**Discussion**

We have used asymmetrically dividing *Drosophila* neuroblasts to provide mechanistic insight into how sibling cell size asymmetry can be established. Understanding the mechanisms underlying the formation of physical asymmetry opens the door to targeted sibling cell size manipulations so that its contribution to cell fate and behavior can be systematically assessed[3]. Sibling cell size asymmetry can be generated through biased cortical expansion, determined through asymmetric Myosin localization; cortical regions containing fewer Myosin filaments will be allowed to expand whereas regions containing high levels of Myosin are prevented to grow[18, 13, 18]. However, the spatiotemporal regulation controlling asymmetric Myosin localization and its dynamics remained elusive.

Here we have shown that two opposing cortical Myosin flows, starting at different times and locations, are a major mechanism to establish asymmetric Myosin distribution. For instance, shortly after anaphase onset, Myosin starts to flow towards the basal cortex, enabling the apical cortex to expand. With a delay of about 1 minute, Myosin subsequently flows from the basal cortex towards the apical pole. This spatiotemporally regulated flow pattern ultimately regulates unequal cortical expansion, necessary for the establishment of physical asymmetry. Our data show that apical Myosin clearing and the delay between the apical and basal Myosin flow onset is regulated through cell cycle and polarity cues.

Cortical flow is triggered through Myosin contractility, pulling Actin filaments and associated proteins towards the contractile Myosin filaments[40]. For cortical flow to start, Myosin contractility would need to be increased or inhibited locally[51]. Here we propose that Myosin flow onset on the basal cortex is induced through a lateral enhancement of Myosin activity, regulated through local delivery of the centralspindlin complex. This model is supported with the following data: (1) the centralspindlin complex component Tumbleweed is accumulating at the lateral neuroblast cortex in a confined position prior to focused lateral Myosin enrichment, which also precedes basal Myosin relocalization. (2) Actomyosin intensified at the lateral neuroblast cortex followed by an increase of activated Myosin. (3) Removal of the mitotic spindle or knocking-down centralspindlin complex components such as *tum*, *pav*, or its upstream regulator *aurB*[20] perturbs basal Myosin clearing. (4) Premature activation of the centralspindlin pathway leads to precocious accumulation of Myosin in the cleavage furrow region, and induces a premature onset of the apically directed Myosin flow.

These results are consistent with a model, proposing that localized activation of the small GTPase Rho1 through the centralspindlin complex results in local Actomyosin filament formation and Myosin activation (Fig. 8g). Furthermore, our laser cutting experiments clearly show that the mitotic spindle is not required to bring Myosin to the cortex but to focus it on the lateral cortex, corresponding to the furrow position. This result is consistent with our earlier observation, showing that the centralspindlin component Pav is already localized at the neuroblast cortex by metaphase (similar to Myosin)[52]. Thus, we conclude that the mitotic neuroblast cortex is primed to bind Myosin filaments already before anaphase but that spindle-dependent cues build up a lateral Myosin gradient, specifically from early anaphase onwards. It is important to note that Myosin flow onset on the apical cortex is independent of both chromatin and spindle cues ([20,52]and the data shown here), but regulated through polarity-induced localization of Protein Kinase N (Pkn) and Rock[31]. Furthermore, theoretical modeling predicted that a gradient of Myosin activity from the poles to the equator is sufficient to induce a cortical flow to the cell equator[53]. However, we currently cannot exclude the existence of cues, modulating Myosin activity on the apical or basal cortex to weaken its contractility and thus enabling the apically or basally directed Myosin flow.

Our results also imply that spindle geometry is an important factor in determining the lateral position of the Myosin gradient, providing an additional layer of regulation, which influences the site of Actomyosin ring formation and subsequent cleavage furrow positioning. Our data are consistent with this notion, showing that compromising either spindle positioning, spindle asymmetry or both have an influence on the site of cleavage furrow formation (Supplementary Fig. 8c) and subsequent physical asymmetry (Supplementary Fig. 8d, e).

Metazoan cells have developed different mechanisms to either prevent or induce sibling cell size asymmetry[33, 54, 55]. In fly neuroblasts, the spatiotemporal regulation of Myosin flow dynamics seems to be necessary and sufficient to induce physical asymmetry; retaining activated Myosin on the apical cortex is causing neuroblasts to invert their physical asymmetry (this study &[31]). Having identified mechanisms to establish physical asymmetry, it will be interesting to test how they will affect cell behavior and fate in fly and other metazoan cells.

## Methods

**Fly strains and genetics.** All mutant chromosomes were balanced over FM7actin::GFP, CyO actin::GFP or TM6B, Tb. The following mutant alleles and RNAi lines were used: pins[P89 56], dlg[m52 57], FRT82B sas4[M 58], rod[H4.8 32], Pav RNAi (v46137; VDRC), Tum RNAi (BL28982; Bloomington; v106850; VDRC), AurB RNAi (VDRC), Sds22 RNAi (IR GD11788)[59], sds22[PB1173 59], Pp1-87B RNAi (v35025; VDRC), Pp1-87B[Bg3] (BL23696; Bloomington), Pp1-87B[Bg6 60], Df(3R)Exel6164 (removes Pp1-87B; Bloomington), Sqh[AX3 61].

**Transgenes and fluorescent markers.** worGal4, UAS-cherry::Jupiter, Sqh::GFP, Histone2A::mRFP1 (Bloomington stock center), UAS-mCherry::CAAX (Bloomington stock center), UAS-Pp1-87B-HA (Bloomington stock center), UAS-Tum::Venus[41], UAS-Ran-Q69L, UAS-Ran-T24N[62], UAS-Gap43::mEos[30], Sqh::mCherry[63], pUAST-attB-ALD-Rock[CA]::VhhGFP4::HA, pUAST-attB-ALD-Rock[CA]::mCherry (this study).

Transgenes were expressed using the neuroblast-specific driver worGal4[64].

Zipper::mDendra2 MiMIC[65] line: Mi02518 was crossed to phiC31 integrase (expressed under the vasa promotor; Bloomington stock center) and the resulting progeny were injected with the mDendra2 exchange cassette[65]. Injections were performed by BestGene. Positive lines were initially screened for loss of yellow body marker and tested for the expression of Zipper::mDendra2.

**Generation of constructs.** Sqh::mDendra2: The mDendra2 coding sequence was PCR amplified and inserted into AscI and NotI restriction sites of attP-Sqh. attP-Sqh was generated by removing EGFP with AscI and NotI. The construct was injected into attP (VK00033 and VK00037).

pUAST-attB-TSmod-Vt: The TSmod-Vt[45] fragment, consisting of mTFP1, the spinder silk protein SSP, Venus and the F-Actin binding domain of Vinculin (Vt) was PCR amplified from the VinTS cDNA obtained from addgene (Plasmid #26019) and subcloned into pUAST-attB between EcoR1 (5′) and Kpn1 (3′) using In-Fusion technology (Takara, Clontech). The construct was inserted at VK00033.

pUAST-attB-ALD-Rock[CA]::VhhGFP4::HA: Inscuteable's apical localization domain (ALD[47]) was PCR amplified and inserted into AscI and MluI restriction sites. Rock's kinase domain and VhhGFP4 fused to HA were PCR amplified and inserted into MluI and XbaI restriction sites using In-Fusion technology (Takara, Clontech). The construct was inserted at VK00033 and VK00037, respectively (Bestgene).

pUAST-attB-ALD-Rock[CA]::mCherry: Inscuteable's apical localization domain (ALD[47]) was PCR amplified and inserted into AscI and MluI restriction sites. Rok's kinase domain and mCherry were PCR amplified and inserted into MluI and XbaI restriction sites using In-Fusion technology (Takara, Clontech). The construct was inserted at VK00033 and VK00037, respectively (Bestgene).

**Antibodies.** The following primary antibodies were used for this study: rat anti-α-Tub (Serotec; 1:500; available from Bio-Rad, Cat#MCA77G), mouse anti-α-Tub (DM1A, Sigma; 1:2500; Cat# T9026), rabbit anti-phospho-Histone 3 (Abcam; 1:1000; Cat# sc-56739), chicken anti-GFP (Abcam; 1:1000; Cat# ab13970), rabbit anti-Ran (Abcam; 1:100; Cat# ab11693), rat anti-Miranda (1:400; gift from Chris Doe), mouse anti-aPKCζ (Santa Cruz; 1:50; discontinued) and guinea pig anti-Sqh1P (1:300)[66]. Secondary antibodies were from Molecular Probes and the Jackson Immuno laboratory.

**Immunostaining.** Ninety-six hours larval brains were dissected in Schneider's insect medium (Sigma-Aldrich S0146) and fixed for 20 min in 4% paraformaldehyde in PEM (100 mM PIPES pH 6.9, 1 mM EGTA and 1 mM MgSO4). After fixing, the brains were washed with PBSBT (1× PBS (pH7,4), 0.1% Triton-X-100 and 1% BSA) and then blocked with 1× PBSBT for 1 h. Primary antibody dilution was prepared in 1X PBSBT and brains were incubated 48 h at 4 °C. Brains were washed with 1× PBSBT four times for 30 min each and then incubated with secondary antibodies diluted in 1X PBSBT at 4 °C, overnight. The next day, brains were washed with 1× PBST (1× PBS, 0.1% Triton-X-100) four times for 20 min each and kept in Vectashield (Vector laboratories) mounting media at 4 °C.

**Live imaging sample preparation.** Imaging medium (Schneider's insect medium (Sigma-Aldrich S0146) mixed with 10% FBS (Sigma), 2% PenStrepNeo (Sigma), 0.02 mg/mL insulin (Sigma), 20mM L-glutamine (Sigma), 0.04 mg/mL L-glutathione (Sigma) and 5 μg/mL 20-hydroxyecdysone (Sigma)) was warmed up to room temperature before use.

Ninety-six hours after egg laying, larval brains were dissected in imaging medium and transferred onto a gas-permeable membrane (YSI Life Sciences 5793) fitted on a metallic slide. Brains were oriented with the brain lobes facing the coverslip. Excess media was removed until the brain lobes were in contact with the coverslip. The sample was sealed with Vaseline. A detailed protocol can be found here[67].

**Primary neuroblast cultures.** For photoconversion experiments, 96 h larval brains were dissected in Chang & Gerhing solution (3.2 g/L NaCl, 3 g/L KCL, 0.69 g/L CaCl₂-₂H2O, 3.7 g/L MgSO₄-7H2O, 1.79 g/L tricine buffer pH 7, 3.6 g/L glucose, 17.1 g/L sucrose, 1 g/L BSA) at room temperature. Brains were then dissociated in Chang & Gerhing solution supplemented in collagenase from Clostridium histolyticum (Sigma) and papain from papaya latex (Sigma) at a final concentration of 1 mg/mL each, during 30 minutes at 30 °C. Brains were washed with imaging medium (see above) and then dissociated in imaging medium by pipetting 20–30 times.

**Imaging.** Fixed samples were imaged using an inverted Leica TSC SPE confocal microscope. For representative images, a 60×/1.40NA oil immersion objective was used. For 4X scans a z-step size of 0.3 μm was used.

Live samples were imaged with an Andor revolution spinning disc confocal system, consisting of a Yokogawa CSU-X1 spinning disk unit and two Andor iXon3 DU-897-BV EMCCD cameras. A 60×/1.4NA oil immersion objective mounted on a Nikon Eclipse Ti microscope was used. Live imaging voxels sizes are 0.22 × 0.22 × 0.5 μm (60x/1.4NA spinning disc).

**Laser cutting experiments.** For laser cutting experiments, Andor's Micropoint system, consisting of a pulsed nitrogen pumped tunable dye laser was used. Ablation was performed using a power of 72%. Imaging was performed before and after ablation using a 60× oil immersion lens (NA 1.4) that was also used to focus the Micropoint laser.

**Photoconversion.** Ninety-six hours larval brains expressing Zipper::mDendra2 were used after their dissociation (see above). The photoconversion experiments were performed on an Andor Revolution spinning disc system containing Andor's FRAPPA unit. Several regions of interests (ROIs) were manually chosen in the GFP channel and Zipper::mDendra2 was irradiated with 405 nm on either the apical,

basal or lateral cortex just after anaphase onset. Before photoconversion, single Z planes containing ROIs were scanned for ten time points with maximum speed. Subsequently, ROIs were irradiated with 405 nm (10%; 50 repeats; 50 μs dwell time). After photoconversion, the entire neuroblast was scanned with a z-step size of 0.65 μm. Converted and unconverted mDendra2 emission were merged in AndorIQ2 and converted into Imaris.

**Fluorescence recovery after photobleaching (FRAP) experiments**. The 488 nm laser line was targeted to region of interests using Andor's FRAPPA module. Images were acquired with high temporal resolution (136ms) after bleaching. Kymographs were generated from the bleached region in Fiji or ImageJ with the Multi Kymograph plugin. Line thickness was set to 3 pixels and lines were drawn on the neuroblast along the cortex (Fig. 2f, Supplementary Fig. 2e) or through the neuroblast, covering both lateral neuroblast regions (Supplementary Fig. 2d). From these kymographs, velocity was extracted by measuring the width (= distance) and height (= time) of the recovery slope. With a pixel size (0.22 μm) and a time resolution (~ 136 ms), the velocity here was calculated with the following formula: Flow velocity (μm/s)=(width in pixel× 0.22)/(height×0.136)

**Colcemid and Flavopiridol experiments**. For colcemid and Flavopiridol experiments, the following strains were used + ; worGal4, UAS-Cherry::Jupiter, Sqh::GFP; + + ; worGal4, UAS-Cherry::Jupiter, Sqh::GFP; rodH4.8 + ; His2A::mRFP1; rod$^{H4.8}$ (this work)

Wild-type or rod$^{H4.8}$ mutant neuroblasts were incubated with colcemid (Sigma) in live imaging medium at a final concentration of 5 μg/mL, or with Flavopiridol hydrochloride (Sigma) at a final concentration of 5 μM. Live imaging was started without delay. Complete spindle depolymerization was seen ~ 30–60 min after colcemid addition.

**Image processing and calculations**. Images were processed using Imaris × 64 7.5.2 and ImageJ. Andor IQ2 files were converted into Imaris files using Imaris File Converter. Measurements of apical and basal Myosin intensity were obtained by using the oblique slicer in Imaris oriented along the apico-basal polarity axis, with a thickness of ~ 3.5 μm. The corresponding image sequences were exported as TIFF files and opened with ImageJ to measure Myosin intensity along the apical and basal cortex. Due to the cellular size which differs from one neuroblast to another, the apical and basal cortical regions used for intensity measurement correspond to ¼ of the diameter of the cell in metaphase. Background corrections were performed by measuring Myosin intensity in the media. Myosin flow velocity was obtained using a custom-made Matlab code. Kymographs, made in ImageJ using the pluggin "MultipleKymograph", were used to analyze the distribution of Sqh and chromosomes. Myosin intensity profiles were established in ImageJ by measuring Sqh::GFP intensities along a line from the apical to the basal cortex (after background correction). Cortical expansion was obtained by measuring the length between the center of the cell in metaphase (used as spatial reference) and the apical/basal cortex in metaphase (=A1) and anaphase (=A2). From these values, the ratio was calculated (A2/A1 and B2/B1, respectively). For laser cutting experiments, Myosin intensity was measured both at the cortex and in the cytoplasm before and after cortical cuts were performed using a line on the entire cortex or a circle in the cytoplasm.

Pictures were cropped in Photoshop and assembled in Illustrator. Quantifications and graphical representations were generated in Microsoft Excel, and Graphpad Prism.

To calculate Myosin intensity ratios (apical/basal), kymographs were generated along the apical-basal polarity axis. On these kymographs, Myosin intensity was measured during anaphase on the apical and basal cortex, respectively. The resulting intensity values were averaged and used to calculate the apical/basal Myosin average intensity ratio.

**Curvature and furrow initiation analysis**. To determine the curvature along the cell cortex, a line was manually drawn in ImageJ from the apical to the basal cortex on the mid-plane. Cortical curvature K can be determined via the following formula: $K = \frac{f''(x)}{\left(1+f'^2(x)\right)^{3/2}}$, where x and f(x) are the horizontal and vertical position of the drawn cortex, respectively. The first and second derivatives (f'(x) and f''(x)) of the curve were calculated numerically using second order difference methods. Custom-written Matlab codes were used to determine curvature values for all points on the curve. To determine furrow initiation, we first determined the average curvature value for the furrow site. Since curvature value at the furrow site will change its sign (from positive to negative or vice versa) when the furrow starts to ingress, the furrow site can be detected by determining the position of the cortex with the highest sign change in the curvature value. Average furrow curvature values were calculated from an average of five nearest points around the peak of the sign change. Furrow initiation (T = 0) is defined as the first time point that the average curvature value changes in sign value. For all time points that furrow ingression was not yet observed, the furrow position at T = 0 was used to determine average value at the furrow.

**Furrow Myosin quantification**. Cell mid-planes were first generated using the Oblique Slicer tool in Imaris (Bitplane) and the entire image volume was then

resliced along the direction of this plane for all time points. Using ImageJ, an average intensity projection was generated from three selected planes closest to the mid-plane. This procedure was done for all acquired time points. To determine cortical intensity signal for both Myosin and polymerized actin markers, a spline curve was drawn along the cell cortex on the average intensity projection image and the XY coordinates of this curve were exported to a text file. Custom made Matlab codes were written to extract the exact XY coordinates of the drawn curve from the text file. Intensity signal of the drawn curve was calculated from the image using an average intensity of the three pixels, closest to the curve. Average furrow Myosin was calculated in the same way as the average curvature described above.

**FRET Imaging and quantifications**. Live samples of cultured neuroblasts were placed in a ibidi 8-well glass bottom 1 μ-slide and imaged on a 3 i spinning disk confocal microscope equipped with a Photometrics Evolve 512 back-illuminated EMCCD camera, using a × 63 1.4 numerical aperture oil-immersion objective. Both donor (mTFP1) and acceptor/FRET (Venus) fluorophores were excited by a diode laser with 440 nm wavelength at 40% power and 200 ms exposure time. Donor and FRET signals were detected using standard CFP (482/35-25) and YFP (542/27-25) emission filter set. The FRET index was determined using custom-written Matlab code. First, a background subtraction was performed for both donor and FRET detected signals using a background averaged noise obtained from 50 different images acquired with the same imaging conditions. Then, the FRET index was calculated by calculating the ratio between the FRET acceptor and FRET donor intensity after subtracting background noise. A cutoff threshold (range 700–1100 a. u.) was used for the donor intensity such that only pixels with intensity above the cutoff are used for FRET index calculations. This cutoff threshold step was required to eliminate artificially high FRET index pixels in the medium due to fluctuating noises. The FRET index was determined for all slices in the z-stack at all acquired time points. To determine average FRET index for the furrow site, a z-focal plane, which best represents the furrow site was selected for every time point for the analysis. The average FRET index around the furrow site was calculated by averaging the FRET index along the cortex with 30 pixels long and 5 pixels thick centered at the furrow site.

**Measurements to determine Myosin and F-Actin intensity changes at the lateral and basal cortex, respectively**. For unbiased and accurate identification of Myosin's lateral enrichment and activity increase (FRET measurements), raw curves were smoothened with the 'loess' method using a smoothing factor of 0.4 to reduce local fluctuations. Local minima, closest to the left of the monotonically rising smoothed curve (i.e., steepest ascending region in the curve) were selected before the signal increased. Local minima and maxima were identified using numerically calculated gradients of smoothed curves. To determine the onset of basal Myosin intensity reduction, we detected the local maxima, closest to the left of the steepest descending region in the smoothened curves. The same method was applied to determine the onset of F-Actin enrichment in the prospective furrow region and for basal F-Actin reduction.

**Definition of statistical tests, sample number, sample collection, replicates**. Statistical significance was calculated using the unpaired samples t test. For each experiment, the data were collected from at least 3 independent experiments. For each independent experiment, at least 5 larvae were dissected.

**Computer codes**. Custom made Matlab codes used for data analysis are available upon request.

**Data availability**. The authors declare that the data supporting the findings of this study are available within the paper and its supplementary information files.

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

## Acknowledgements

We thank members of the Cabernard lab for helpful discussions and Emmanuel Gallaud, François Payre and Linda Wordeman for critical reading of the manuscript. We are also grateful to Barry Thompson and David Glover for flies and antibodies. We would also like to thank the Imaging Core Facility (IMCF) at the Biozentrum for technical support, Damian Brunner and Werner Boll for providing access to their laser ablation system and the Nigg lab for providing temporary lab space to C.R. This work was supported through funding from the Swiss Initiative in Systems Biology (SystemsX; SXFSIO_141991, IPhD 51PHP0_157299 and MorphogenetiX), the Swiss National Science Foundation (SNSF; PP00P3_133658, PP00P3_159318, 310030_156836), Worldwide Cancer Research (14–0236), funds from the Kanton Basel-L and and Basel-Stadt, and start-up funds from the University of Washington. C.R. was supported with an EMBO long-term post-doctoral fellowship. Stocks obtained from the Bloomington *Drosophila* Stock Center (NIH P40OD018537) and the Vienna *Drosophila* Resource Center (VDRC) were used in this study.

## Author contributions

The majority of this study was conceived by C.R. and C.C., and C.R. performed most of the experiments. A.T. and C.C.: Conceived, designed and performed the initial Myo:: mDendra2 experiments, resulting in some of the main conclusions. T.T.P.: Performed the F-Actin localization, FRET and curvature experiments, and wrote custom-made Matlab codes for data analysis. A.M.: Performed the FRAP experiments. The nanobody technology was conceived by E.C. and M.A. All authors contributed to data interpretation. C. R. and C.C. wrote the paper with help from T.T.P.

## Additional information

**Competing interests:** The authors declare no competing financial interests.

