## [Peer Review File · Nature Communications]

Reviewers' Comments:

Reviewer #1 (Remarks to the Author)

In this paper, Cabenard and colleagues combine genetic and pharmacological manipulations with photokinetics to explore how cleavage furrow placement and physical asymmetry emerges through spatiotemporal control over Myosin II localization during cytokinesis. They provide evidence for two flows: An apical -> basal flow directed by a previously described apical polarity pathway, and a basal -> equator flow that they argue here is directed by spindle position via equatorial activation of centralspindlin, which produces a gradient of myosin activity. They argue that basal myosin clearing is caused primarily by the basal -> equator flow, and they show that the physical size asymmetry of daughter cells produced at cleavage is strongly correlated with the relative timing of apical and basal flows, although spindle position and asymmetry make a smaller contribution when the timing of apical and basal flows are equalized.

In general, the paper is well-written and the data are clean and presented clearly. There is some valuable new information about how multiple pathways (polarity and centralspindlin) and mechanisms (flow and local recruitment) are integrated to choreograph a complex spatiotemporal pattern of Myosin II localization to effect control cleavage furrow position and daughter cell size asymmetry. However, I have some concerns about the extent to which some of the central claims of the paper are supported by the data. I expect that the majority of these concerns can be addressed by further analysis of existing data or by a more nuanced interpretation of some of the observations.

Major comments

(1) A major point of the paper is that cortical flows play a major role in determining myosin II distribution, but much of the evidence supporting this claim is fairly weak. For example, the photoconversion (of Myosin II) results shown in Figure 1 are consistent with redistribution of Myosin II by cortical flow, but they are also consistent with a scenario in which there is spatiotemporal patterning of Myosin II activation and turnover, with net disassembly/dissociation at poles and net activation/assembly at equator. The control experiment of activating myosin laterally and showing that it remains there does not really distinguish these two scenarios. The compaction of the lateral activation spot over time is consistent with local compressive flow, but this says nothing about the quantitative contribution that flow makes to relocalizing Myosin, relative to other mechanisms. The data shown in Figure 6b is more convincing, because it shows clear translocation of a photoconverted patch. Indeed, until I saw this data, I was not really convinced that there was significant flow. Perhaps these data should be shown much earlier....

(2) In Fig 3c, the bulk of equatorial accumulation occurs before basal clearing, so how can relocalization by flow make a major contribution to equatorial accumulation?

(3) Clearly Myosin II is patterned by some combination of local inputs that bias activation/inactivation (polarity, centralspindlin, etc) and redistribution by cortical flow. Given this, I think it is not sufficient to document that flows occur and that there is rough correlation between flow and redistribution, and then claim that flows make the major contribution. I think the authors need to make a more quantitative argument about the magnitude of the contribution. Loosely speaking, the magnitude should depend on the rate of flow, and the cortical lifetime of Myosin II

because this determines how far a molecule of Myosin II will be transported before it falls off again. I think the photokinetic experiments can provide estimates of both of these quantities (through estimates of velocity and decay in intensity of the photoactivated patch). My feeling, looking at the data in Figure 6b, is that the contribution is significant, but I think the argument needs to be made more carefully and quantitatively, because this is a central point of the paper.

(4) The basic argument that equatorial activation of Myosin II creates a gradient of contractility that drives a flow of cortex from basal to apical is problematic. Equatorial Myosin activation occurs against a global asymmetry in which there is far less Myosin II apically than basally. Intuitively, I would expect that imposing an equatorial enrichment of Myosin II over a apico basal asymmetry would cause the equatorial cortex to contract most strongly, but that this would be imposed on a contraction of the entire basal cortex away from the apical pole. I don't see how to explain the observed pattern of cortical flow without either: (a) weakening the basal cortex or (b) providing some additional mechanism to anchor the equatorial cortex position against basal contraction.

(5) Related to this, the authors mention that in mutant cells lacking the mitotic spindle and components of CP or Centralspindlin, that basal clearing still occurs, albeit with a delay. It seems to me that a more realistic model for the control of basal - equator cortical flows is that it requires a combination of equatorial Myosin accumulation and basal inhibition/disassembly.

(6) The F-actin experiments reported in Figure 4 are particularly weak. First of all, it is possible/likely that F-actin assembly is stimulated at the equator by RhoA activation, so How can observed increase be attributed to flow without doing a control experiment to show that the accumulation does not occur when flow is blocked? Second, I do not understand how the authors can pinpoint the timing of an increase in F-actin "crowding" from the kind of data shown in Figure 4, so any arguments about relative timing of Myosin accumulation, flow and actin crowding seem very suspicious.

(7) I find the author's interpretation of the laser cutting experiments very confusing. What does it mean exactly to say that the spindle is required to "confine" Myosin II? This makes it sound like they are saying that it is required to prevent Myosin II from moving away. What the experiment shows (to me at least) is that the pattern of Myosin return following laser cuts depends on the spindle. It is not clear to what extent this pattern reflects contributions from local recruitment, flow, or other things.

Minor comments

Throughout the paper, the authors refer to events occurring during metaphase or anaphase or telophase, but in all of the timetraces, only the onset of furrowing is shown as a temporal point of reference. It would be helpful to include a simple schematic up front for non-specialist readers showing the relationship between the events discussed here (apical and basal clearing and flows, etc and cell cycle phases).

(2) Figure 3a,b: Why is the localization of Myosin II so much broader than tum? Is the pattern of RhoA activation broader?

(3) Figure 6: In pins and pins;dlg mutants, they report the relative delay between apical and basal

flows reported. What is the absolute delay?

Reviewer #2 (Remarks to the Author)

The manuscript by Roubinet et al., uses *Drosophila* neuroblasts as a model to study asymmetric cell division, an important process during organismal development and tissue regeneration. This manuscript focuses on the aspect of unequal daughter cell size control. In a series of elegant live cell imaging and sophisticated manipulation experiments, the authors demonstrate that two temporally distinct myosin flows contribute to the positioning of the myosin contractile zone at the lateral cortex of neuroblasts. The authors hypothesize and provide supporting data for a model that suggests that the difference in timing of the two cortical flows during anaphase together with spindle-dependent cues established an asymmetric positioning of the cleavage plane. Furthermore, the authors suggest that the centralspindlin-dependent pathway eliciting contractility during cytokinesis is implicated in the 2nd flow.

The manuscript is well written and interesting to read. The quality and quantitative analysis of time-lapse experiments and manipulations is excellent. One criticism is that the experiments in the manuscript, while very well executed, provide little mechanistic insights regarding the processes underlying and spatiotemporally controlling the two myosin flows. The main specific points (listed below) provide some suggestions of how to potentially strengthen this part of the work.

The work provides new insights into the important process of asymmetric cell division and will be interesting to a wide readership in different areas of biomedicine. Providing additional mechanistic insights would increase the impact of the work and make it a strong candidate for publication in *Nature Communications*.

Specific points:

- (1) The authors mention that myosin flows involve myosin activity. This should be tested using conditional myosin mutants or small molecule inhibitors of myosin II ATPase activity. Acute and reversible myosin inhibition may also allow the authors to manipulate the time window between apical and basal clearance and thereby establish a more causal role for the window in establishing daughter cell size asymmetry.
- (2) The first myosin movement leads to the clearance of myosin from the apical cortex. The second wave leads to the basal clearance and lateral relocalization of myosin. What is the molecular basis for this temporal separation? The temporal difference appears critical based on the model presented in the manuscript. Is a polarized inactivation of CDK/CyclinB responsible for this difference during mitotic exit (earlier on apical or basal side)? This could be tested by recording GFP fusions of cyclinB or by staining using a generic phosphoCDK substrate antibody. Have the authors explored other potential mechanisms underlying this temporal separation?
- (3) The authors suggest that the contractility promoting function of centralspindlin at the lateral cortex helps to remove basal myosin. The key effect of centralspindlin in cytokinesis is thought to be the RhoGEF ECT2/Pebble. If the model proposed is correct, pebble mutant or depleted cells should also show a defect in basal myosin flow to the lateral side. This should be tested.
- (4) Lastly, while the authors provide some mechanistic insights into the basal to lateral flow, the mechanism of the earlier apical to lateral flow remains unaddressed in this work. The authors mention that this will be presented in another paper. However, I am not entirely sure it is helpful to separate this and present it in two different papers.

Reviewer #3 (Remarks to the Author)

In Roubinet et al the question of how differently sized cells arise from a single division is addressed. This is an important yet relatively unexplored topic in cell and developmental biology. The study uses *Drosophila* neuroblasts as a model focusing on the role of Myosin II at the cortex

during anaphase. According to the manuscript two cortical flows one in an apical-to-basal direction occurring first, followed with a delay by a basal-to-apical directed flow redistribute Myosin II thereby influencing final furrow positioning and where cleavage will take place. This process is independent of chromatin-mediated cues but influenced by signals from the mitotic spindle. It is the delay between the two flows that was identified to be a critical determinant of final furrow positioning. The approaches chosen are original (i.e. photo-conversion of Myosin to measure flows and laser cutting to measure Myosin dynamics during the process). The findings - if proven to be true - would represent a significant advance in our understanding of the process.

However, important conclusions are currently not backed up by experimental evidence. The manuscript lacks clarity and basic elements i.e. indication of sample size are missing for many experiments. If the authors can address the issues below and in case that the new results convincingly demonstrate what the paper claims a revision could be considered.

Most important issues:

In line 128 the conclusion is made that the cortical flow is a major mechanism to re-localize Myosin from both poles. I don't think the data demonstrate that for the apical pole. It must be excluded that Myosin is recruited from the cytoplasm to demonstrate this first flow repeating the logic of figure 1f (i.e. convert one half of the apical Myosin pool). If converted signal accumulates only on one half of the furrow, this would be a strong argument. Without that how can it be ruled out that the observed effect is not caused by Myosin losing competence to bind the apical cortex? After the laser cut induced membrane "trauma", Myosin is uniformly recruited in metaphase cuts, but is not recruited to the apical pole in anaphase cuts (fig 5) consistent with that alternative. Fig 5b shows that Myosin can be recruited to the furrow in high levels from the cytoplasm questioning the need for a flow.

The spindle sends a signal via Tumbleweed. In that regard I fail to link up the line scans in figure 3b with the microscopic image. At 30s cortical Tum signal at the "equator" of the cell can be seen, yet the line scan does not pick that up. Assuming the scan is correct, there is clearly Myosin enrichment between 0s and 30s, while Tum scan signal is unchanged. This does not fit the concluded temporal order of events.

Figure 3d-g would make a strong point if for each condition the average intensity of several neuroblasts would be used to generate the 0s, 60s and 90s panels to make this quantitative.

The FRET data are not acceptable. This is a new tool that must be demonstrated to actually report what it is supposed to measure. Assuming the reporter to be fine, it does not help that the sensor is referred to as intramolecular (line 410) as well as intermolecular (line 200). Figure 3 shows the "beginning" of the events but an objective standard to measure this is missing. For the FRET sensor the data are plotted in the range of -50 to -20 (fig 4e). However the graph (fig 4c) shows a massive peak -100 to -60. I can't believe given the very sensitive nature of FRET that this peak is irrelevant. A further problem here is that there is something wrong in the annotation of the timing between Fig 4 b and c. I assume the measures are taken on the same neuroblast since the curvature graphs are identical.

The correlation of the delay of flows and final furrow position in figure 6 is intriguing. However, the correlation for the blue WT dots is less convincing which might be caused by the low sample size (9). They say that in wild type neuroblasts the delay between the 2 flows is $65.56s \pm 13.33$ (line 243) this is ~20% of variation! This should be enough to establish a correlation if the model is correct. Measuring the delay in a large number of wild type neuroblasts (30-50?) should be easily doable. To make this solid, accurate daughter cell size measurements should be provided (e.g. following Homem et al. 2013). While I agree, the data in fig 6e are very likely to reflect daughter

cell size differences they do not formally show that. This would also help to replace the somewhat obscure term of "physical asymmetry" when they try to understand daughter cell size differences. Mud mutant neuroblasts should be included in this analysis.

The effect of changes in the geometry of the spindle apparatus on daughter cell size asymmetry has been observed before. This needs to be acknowledged and discussed (e.g. Cai et al Cell, 03, Yu et al JCB 03).

A table showing the sample size for each experiment must be provided.

Other important issues

Line 184. "Myosin enrichment at the lateral cortex". It needs to be explained precisely what they mean here by "Myosin enrichment on the lateral cortex". Enriched compared to what? It is already enriched at the lateral cortex compared to the apical pole at 0s, it's enriched at the lateral pole at 30s vs 0s, and it's already enriched at the lateral pole compared to the basal pole at 60s.

Measurements for the delay should be shown for sds22 to demonstrate it's not involved in clearing.

Figure 2c has to be rod+colc, not colcemid as panel and legend suggest. Without rod clearing of basal Myosin cannot occur.

Fig 5c – how can you possibly determine anaphase in rod colc neuroblasts?

Line 244: On what basis is apical determined in dlG;pins (check nomenclature) neuroblasts that express Myosin? larger = apical? This would be a circular argument.

Given the nature of the dataset used for Myosin speed measurements their value is questionable.

There is a typo in the title.

Line 287 should refer to Fig7.

Line 416 check color annotation.

Reviewers' comments:

Reviewer #1 (Remarks to the Author):

In this paper, Cabenard and colleagues combine genetic and pharmacological manipulations with photokinetics to explore how cleavage furrow placement and physical asymmetry emerges through spatiotemporal control over Myosin II localization during cytokinesis. They provide evidence for two flows: An apical -> basal flow directed by a previously described apical polarity pathway, and a basal -> equator flow that they argue here is directed by spindle position via equatorial activation of centralspindlin, which produces a gradient of myosin activity. They argue that basal myosin clearing is caused primarily by the basal -> equator flow, and they show that the physical size asymmetry of daughter cells produced at cleavage is strongly correlated with the relative timing of apical and basal flows, although spindle position and asymmetry make a smaller contribution when the timing of apical and basal flows are equalized.

In general, the paper is well-written and the data are clean and presented clearly. There is some valuable new information about how multiple pathways (polarity and centralspindlin) and mechanisms (flow and local recruitment) are integrated to choreograph a complex spatiotemporal pattern of Myosin II localization to effect control cleavage furrow position and daughter cell size asymmetry. However, I have some concerns about the extent to which some of the central claims of the paper are supported by the data. I expect that the majority of these concerns can be addressed by further analysis of existing data or by a more nuanced interpretation of some of the observations.

Major comments

(1) A major point of the paper is that cortical flows play a major role in determining myosin II distribution, but much of the evidence supporting this claim is fairly weak. For example, the photoconversion (of Myosin II) results shown in Figure 1 are consistent with redistribution of Myosin II by cortical flow, but they are also consistent with a scenario in which there is spatiotemporal patterning of Myosin II activation and turnover, with net disassembly/dissociation at poles and net activation/assembly at equator. The control experiment of activating myosin laterally and showing that it remains there does not really distinguish these two scenarios. The compaction of the lateral activation spot over time is consistent with local compressive flow, but this says nothing about the quantitative contribution that flow makes to relocalizing Myosin, relative to other mechanisms. The data shown in Figure 6b is more convincing, because it shows clear translocation of a photoconverted patch. Indeed, until I saw this data, I was not really convinced that there was significant flow. Perhaps these data should be shown much earlier....

We have consolidated all information related to Myosin flow in a new Figure 2. In addition to the existing photoconversion experiments, we performed FRAP experiments at the lateral cortex prior to flow onset in metaphase and later – when flow should be well underway – in anaphase. We reasoned that if flow occurs, Myosin should fill in the bleached region from the edges of the bleached region. Alternatively, if Myosin turnover is a predominant mechanism for its redistribution, we would have expected that the bleached region fills in with no particular

directionality. Kymograph analysis reveals a “V” shape pattern, with Myosin filling in the bleached region from the edges and thus with clear directionality. However, this pattern is only observed in anaphase but not in metaphase. We quantified the flow velocity from these kymographs and obtained very similar values as shown for the photoconversion data.

We propose that Myosin flow is a major – not the sole – mechanism for its relocalization.

(2) In Fig 3c, the bulk of equatorial accumulation occurs before basal clearing, so how can relocalization by flow make a major contribution to equatorial accumulation?

We reanalyzed this data and also measured apical flow dynamics in single cells, imaged with high temporal resolution. This new data is shown in Figure 1d. Consistent with the photoconversion data, showing that apically and basally photoconverted Myosin reaches the cleavage furrow, we show in Figure 1d that Myosin relocalization follows a very clear pattern: (1) apical clearing occurs first, (2) equatorial enrichment is next and (3) basal depletion/clearing ensues. Thus, the initial equatorial enrichment is most likely due to Myosin filaments originating from the apical cortex.

(3) Clearly Myosin II is patterned by some combination of local inputs that bias activation/inactivation (polarity, centralspindlin, etc) and redistribution by cortical flow. Given this, I think it is not sufficient to document that flows occur and that there is rough correlation between flow and redistribution, and then claim that flows make the major contribution. I think the authors need to make a more quantitative argument about the magnitude of the contribution. Loosely speaking, the magnitude should depend on the rate of flow, and the cortical lifetime of Myosin II because this determines how far a molecule of Myosin II will be transported before it falls off again. I think the photokinetic experiments can provide estimates of both of these quantities (through estimates of velocity and decay in intensity of the photoactivated patch). My feeling, looking at the data in Figure 6b, is that the contribution is significant, but I think the argument needs to be made more carefully and quantitatively, because this is a central point of the paper.

Thank you for this insightful suggestion. We spent a lot of time trying to figure out how we can measure cortical lifetime either with existing data or with new experiments. We think that the photoconversion data lend itself very poorly to extract cortical lifetime because the photoconvertible mDendra2 is not very photostable in our hands. Thus, photobleaching would significantly negatively affect the quality of the data. Instead, we tried to obtain TIRF data, by imaging Myosin::GFP in isolated neuroblasts. However, TIRF data can only be obtained if the cells are flattened to a certain degree, which we achieved with different coating conditions. Unfortunately, the coating negatively affected cell division (many cells failed to complete cytokinesis) and we thus are not confident that the obtained data accurately reflect Myosin dynamics. Although TIRF might be a viable option sometime in the future, we need to invest more time in finding ideal coating conditions and ways to analyze the data quantitatively to extract cortical lifetime data of high quality.

(4) The basic argument that equatorial activation of Myosin II creates a gradient of contractility

that drives a flow of cortex from basal to apical is problematic. Equatorial Myosin activation occurs against a global asymmetry in which there is far less Myosin II apically than basally. Intuitively, I would expect that imposing an equatorial enrichment of Myosin II over a apico basal asymmetry would cause the equatorial cortex to contract most strongly, but that this would be imposed on a contraction of the entire basal cortex away from the apical pole. I don't see how to explain the observed pattern of cortical flow without either: (a) weakening the basal cortex or (b) providing some additional mechanism to anchor the equatorial cortex position against basal contraction.

Based on previously published simulations and our experimental data presented here we think that equatorial Myosin activation could provide the motor for the apical-directed actomyosin flow. However, we do not exclude that additional signals are needed to modify Myosin contractility – counteracting the apically directed flow - on the basal cortex. We include a sentence in the discussion saying that at this point, we cannot exclude the possibility that local signaling cues might be necessary to weaken Myosin activity on the basal cortex. Our data however is consistent with a model suggesting that equatorial Myosin activation could provide the motor for an apically directed Myosin flow.

(5) Related to this, the authors mention that in mutant cells lacking the mitotic spindle and components of CP or Centralspindlin, that basal clearing still occurs, albeit with a delay. It seems to me that a more realistic model for the control of basal - equator cortical flows is that it requires a combination of equatorial Myosin accumulation and basal inhibition/disassembly.

We don't exclude this possibility – as mentioned in the discussion but currently, we don't have mechanistic insight into basal Myosin inhibition/disassembly mechanisms.

(6) The F-actin experiments reported in Figure 4 are particularly weak. First of all, it is possible/likely that F-actin assembly is stimulated at the equator by RhoA activation, so How can observed increase be attributed to flow without doing a control experiment to show that the accumulation does not occur when flow is blocked? Second, I do not understand how the authors can pinpoint the timing of an increase in F-actin “crowding” from the kind of data shown in Figure 4, so any arguments about relative timing of Myosin accumulation, flow and actin crowding seem very suspicious.

This is a fair point and we completely revised this figure with new experiments and a new sensor. First, we co-imaged Myosin (Sqh::mCherry) with F-Actin (lifeact) in wild type neuroblasts and measured Myosin and F-Actin accumulation in the prospective furrow region and its concomitant clearing on the basal cortex. In addition, we used a Myosin activity sensor, using two vinculin domains separated by a FRET module and a flexible spider silk protein (Grashoff et al., *Nature*, 466(7303), 263–266) as a tool to detect the increase in activated Myosin in the furrow region. All these datasets are correlated with changes in curvature in the furrow region. So, the timing is always in reference to the start of furrow ingression. This data shows quite clearly that both Myosin and F-Actin accumulate in the prospective furrow region well before furrowing commences. It also shows that after this accumulation of Myosin and F-Actin, Myosin activity starts to increase. Finally, our measurements show that Myosin and F-Actin

clear on the basal cortex only after this initial equatorial accumulation and increase in activity is detected.

In sum, we improved the measurements and thus the quality of the data to better support our model.

(7) I find the author's interpretation of the laser cutting experiments very confusing. What does it mean exactly to say that the spindle is required to "confine" Myosin II? This makes it sound like they are saying that it is required to prevent Myosin II from moving away. What the experiment shows (to me at least) is that the pattern of Myosin return following laser cuts depends on the spindle. It is not clear to what extent this pattern reflects contributions from local recruitment, flow, or other things.

Yes. We also conclude that the pattern of Myosin return – not the return itself – is spindle-dependent. This fits with our model suggesting that Myosin enrichment in the furrow region is dependent on the mitotic spindle. We reformulated this in the text and state that the mitotic spindle is necessary for "focused" Myosin localization.

Minor comments

Throughout the paper, the authors refer to events occurring during metaphase or anaphase or telophase, but in all of the timetraces, only the onset of furrowing is shown as a temporal point of reference. It would be helpful to include a simple schematic up front for non-specialist readers showing the relationship between the events discussed here (apical and basal clearing and flows, etc and cell cycle phases).

We provided better annotations of Figure 1a, providing an overview of Myosin relocalization dynamics during the neuroblast cell cycle and also added a schematic cartoon in supplemental figure 1.

(2) Figure 3a,b: Why is the localization of Myosin II so much broader than tum? Is the pattern of RhoA activation broader?

Unpublished observations suggests that RhoA localization is very similar to Myosin II localization. However, we do not yet know whether this corresponds to activated RhoA.

(3) Figure 6: In pins and pins;dlg mutants, they report the relative delay between apical and basal flows reported. What is the absolute delay?

We have added this data to Figure 1.

Reviewer #2 (Remarks to the Author):

The manuscript by Roubinet et al., uses *Drosophila* neuroblasts as a model to study asymmetric cell division, an important process during organismal development and tissue regeneration. This manuscript focuses on the aspect of unequal daughter cell size control. In a series of elegant live cell imaging and sophisticated manipulation experiments, the authors demonstrate that two temporally distinct myosin flows contribute to the positioning of the myosin contractile zone at the lateral cortex of neuroblasts. The authors hypothesize and provide supporting data for a model that suggests that the difference in timing of the two cortical flows during anaphase together with spindle-dependent cues established an asymmetric positioning of the cleavage plane. Furthermore, the authors suggest that the centralspindlin-dependent pathway eliciting contractility during cytokinesis is implicated in the 2nd flow.

The manuscript is well written and interesting to read. The quality and quantitative analysis of time-lapse experiments and manipulations is excellent. One criticism is that the experiments in the manuscript, while very well executed, provide little mechanistic insights regarding the processes underlying and spatiotemporally controlling the two myosin flows. The main specific points (listed below) provide some suggestions of how to potentially strengthen this part of the work.

The work provides new insights into the important process of asymmetric cell division and will be interesting to a wide readership in different areas of biomedicine. Providing additional mechanistic insights would increase the impact of the work and make it a strong candidate for publication in *Nature Communications*.

Specific points:

(1) The authors mention that myosin flows involve myosin activity. This should be tested using conditional myosin mutants or small molecule inhibitors of myosin II ATPase activity. Acute and reversible myosin inhibition may also allow the authors to manipulate the time window between apical and basal clearance and thereby establish a more causal role for the window in establishing daughter cell size asymmetry.

Thank you for this insightful comment. We have addressed this with a new experiment, using a single chain antibody – called a nanobody (vhhGFP4) - that has a high specificity for GFP. We further fused vhhGFP4 with the kinase domain of Rho kinase and also added an apical localization domain. With this tool, we managed to bias Myosin activity specifically on the apical cortex, preventing Myosin clearing. Consistent with our model, this experiment showed that neuroblasts reverted sibling cell size asymmetry, producing a small apical and a large basal cell. This new data is shown in Figure 7d-f and supplemental figure 7b-d.

(2) The first myosin movement leads to the clearance of myosin from the apical cortex. The second wave leads to the basal clearance and lateral relocalization of myosin. What is the molecular basis for this temporal separation? The temporal difference appears critical based on the model presented in the manuscript. Is a polarized inactivation of CDK/CyclinB responsible for this difference during mitotic exit (earlier on apical or basal side)? This could be tested by recording GFP fusions of cyclinB or by staining using a generic phosphoCDK substrate

antibody. Have the authors explored other potential mechanisms underlying this temporal separation?

Previous data showed that CyclinB showed no asymmetric/biased localization in fly neuroblasts (see Figure 4a in Caous et al., Nature Communications 2015).

However, we used a CDK1 inhibitor (flavopiridol) and showed that neuroblasts initiate apical clearing prematurely compared to wild type neuroblasts. Thus, the initiation of apical Myosin relocalization is cell cycle dependent. Furthermore, we also found that the temporal difference between apical and basal Myosin relocalization also responds to both cell polarity – it is reduced in *dlg;;pins* or *pins* single mutants – and the cell cycle because flavopiridol treated neuroblasts also compromise this window. These results are consistent with previous data showing that cell polarity is linked with the cell cycle in fly neuroblasts. This data is added to Figure 1.

(3) The authors suggest that the contractility promoting function of centralspindlin at the lateral cortex helps to remove basal myosin. The key effect of centralspindlin in cytokinesis is thought to be the RhoGEF ECT2/Pebble. If the model proposed is correct, pebble mutant or depleted cells should also show a defect in basal myosin flow to the lateral side. This should be tested.

We knocked-down pebble using inducible RNAi and found the same phenotype as knock-down of centralspindlin (*pav* or *tum*). This data is shown in Figure 4f.

(4) Lastly, while the authors provide some mechanistic insights into the basal to lateral flow, the mechanism of the earlier apical to lateral flow remains unaddressed in this work. The authors mention that this will be presented in another paper. However, I am not entirely sure it is helpful to separate this and present it in two different papers.

In the revised manuscript, we provide more data showing that cell cycle and polarity cues are implicated in the initiation of the apical flow. However, due to the amount of data already presented here we decided to split the results.

Reviewer #3 (Remarks to the Author):

In Roubinet et al the question of how differently sized cells arise from a single division is addressed. This is an important yet relatively unexplored topic in cell and developmental biology. The study uses *Drosophila* neuroblasts as a model focusing on the role of Myosin II at the cortex during anaphase. According to the manuscript two cortical flows one in an apical-to-basal direction occurring first, followed with a delay by a basal-to-apical directed flow redistribute Myosin II thereby influencing final furrow positioning and where cleavage will take place. This process is independent of chromatin-mediated cues but influenced by signals from the mitotic spindle. It is the delay between the two flows that was identified to be a critical determinant of final furrow positioning. The approaches chosen are original (i.e. photo-conversion of Myosin to measure flows and laser cutting to measure Myosin dynamics during the process). The findings - if proven to be true - would represent

a significant advance in our understanding of the process.

However, important conclusions are currently not backed up by experimental evidence. The manuscript lacks clarity and basic elements i.e. indication of sample size are missing for many experiments. If the authors can address the issues below and in case that the new results convincingly demonstrate what the paper claims a revision could be considered.

Most important issues:

In line 128 the conclusion is made that the cortical flow is a major mechanism to re-localize Myosin from both poles. I don't think the data demonstrate that for the apical pole. It must be excluded that Myosin is recruited from the cytoplasm to demonstrate this first flow repeating the logic of figure 1f (i.e. convert one half of the apical Myosin pool). If converted signal accumulates only on one half of the furrow, this would be a strong argument. Without that how can it be ruled out that the observed effect is not caused by Myosin losing competence to bind the apical cortex? After the laser cut induced membrane "trauma", Myosin is uniformly recruited in metaphase cuts, but is not recruited to the apical pole in anaphase cuts (fig 5) consistent with that alternative. Fig 5b shows that Myosin can be recruited to the furrow in high levels from the cytoplasm questioning the need for a flow.

The experiment outlined above is essentially shown in the new Figure 2d (top half; formerly Figure 6b). This is a representative neuroblast where Myosin was converted very close to the tip of the apical cortex. This is the closest we can get. We cannot exclude that the initiating event is a local downregulation of Myosin prior to flow onset. However, our data are in support of a model proposing that both basally and apically directed Myosin flows exist. To further support this model, we performed FRAP experiments to measure whether Myosin fills in the bleached region with a particular directionality. Indeed, in anaphase Myosin recovers from the edges first, which is not the case in metaphase.

The spindle sends a signal via Tumbleweed. In that regard I fail to link up the line scans in figure 3b with the microscopic image. At 30s cortical Tum signal at the "equator" of the cell can be seen, yet the line scan does not pick that up. Assuming the scan is correct, there is clearly Myosin enrichment between 0s and 30s, while Tum scan signal is unchanged. This does not fit the concluded temporal order of events.

Tumbleweed has not yet reached the equatorial cortex but is localized at the very tips of microtubules contacting the equatorial cortex (such as shown in Figure 4a). Since the profiles have been done on the cortex, excluding any cytoplasmic signal (including microtubule tips), the Tum intensity profiles don't yet pick this up.

We agree that Myosin is already at the cortex between 0s and 30s and cleared on the apical cortex. However, Myosin does not show focused localization at the lateral cortex where the cleavage furrow will form. Myosin profiles at 0s and 30s show no difference between equatorial and basal cortex. Myosin shows focused lateral enrichment at 90s. This enrichment overlaps with Tum.

Figure 3d-g would make a strong point if for each condition the average intensity of several neuroblasts would be used to generate the 0s, 60s and 90s panels to make this quantitative.

We improved this dataset by averaging several neuroblasts (n=5) for wild type, *rod* mutant colcemid-treated neuroblasts, Tumbleweed, Pebble and Pavarotti RNAi. Average intensity and standard deviation are shown now.

The FRET data are not acceptable. This is a new tool that must be demonstrated to actually report what it is supposed to measure. Assuming the reporter to be fine, it does not help that the sensor is referred to as intramolecular (line 410) as well as intermolecular (line 200). Figure 3 shows the “beginning” of the events but an objective standard to measure this is missing. For the FRET sensor the data are plotted in the range of -50 to -20 (fig 4e). However the graph (fig 4c) shows a massive peak -100 to -60. I can't believe given the very sensitive nature of FRET that this peak is irrelevant. A further problem here is that there is something wrong in the annotation of the timing between Fig4 b and c. I assume the measures are taken on the same neuroblast since the curvature graphs are identical.

We completely revised this figure and measured both Myosin and F-Actin intensity at the prospective cleavage furrow and the basal cortex in relation to furrow ingression (curvature). In addition, we implemented a novel sensor, measuring Myosin activity by responding with high FRET when F-Actin filaments are being pulled together. The rational and controls for this sensor are shown in supplemental Figure 5. Data derived from this sensor supports the conclusion that Myosin activity (we don't refer to F-Actin crowing anymore) increases already prior to furrow ingression. Neuroblasts treated with the Rho kinase inhibitor do not show such an increase.

The correlation of the delay of flows and final furrow position in figure 6 is intriguing. However, the correlation for the blue WT dots is less convincing which might be caused by the low sample size (9). They say that in wild type neuroblasts the delay between the 2 flows is $65.56s \pm 13.33$ (line 243) this is ~20% of variation! This should be enough to establish a correlation if the model is correct. Measuring the delay in a large number of wild type neuroblasts (30-50?) should be easily doable. To make this solid, accurate daughter cell size measurements should be provided (e.g. following Homem et al. 2013). While I agree, the data in fig 6e are very likely to reflect daughter cell size differences they do not formally show that. This would also help to replace the somewhat obscure term of “physical asymmetry” when they try to understand daughter cell size differences. *Mud* mutant neuroblasts should be included in this analysis.

We have measured a large number of wild type neuroblasts (n = 50) and could establish the correlation between Myosin relocalization timing and sibling cell size ratios. This new data is shown in supplemental Figure 7a. Instead of *mud* mutant neuroblasts, which show a more complex phenotype, we used nanobody technology to bias Myosin activity to the apical cortex. For instance, we used the previously published vhhGFP4 nanobody and fused it to the kinase domain of Rho Kinase. This construct was combined with *inscuteable*'s apical localization domain, to bias Myosin activity on the apical neuroblast cortex. This construct prevents or reduces apical Myosin relocalization, causing sibling cell size asymmetry to invert.

Measuring daughter cell size accurately is very difficult to do. It would require to segment both sibling cells so that their volume can be measured. However, this is only possible with a uniform and bright membrane marker, which we don't have in this experiment. Also, data would have to be required with very thin spacing along the Z axis. In sum, our data is not suitable for this type of analysis. In the Homem et al. paper, nuclear size was used to measure cell size after cell division. So, they did not measure sibling cell size asymmetry but cell growth. This is how they wrote it:

“To compare type I and type II NB sizes, the nuclear volumes during the 9 min preceding mitosis were averaged as nuclei reached their maximal size during this time. Nuclear volumes of INPs were determined as the average volume during the 39-45 min after the INP could first be detected. Nuclear volumes of GMCs were determined as the average volume during the 9-15 min after the GMC could first be detected. Growth rates were defined as the inverse ratio between average volumes over 9min after mitosis and 9 min before the next mitosis.”

Thus, we think that their method cannot be applied in our case. Given the difficulties of segmenting cells during cell division, we decided measuring the distance between the apical or basal cortex and the cleavage furrow is a more direct measurement of the asymmetry than to average nuclei size before and after mitosis.

The effect of changes in the geometry of the spindle apparatus on daughter cell size asymmetry has been observed before. This needs to be acknowledged and discussed (e.g. Cai et al Cell, 03, Yu et al JCB 03).

We modified the manuscript accordingly.

A table showing the sample size for each experiment must be provided.

We submitted this table for the reviewer. Since we already show the sample size for all experiments, we think it is unnecessary to include this table as supplemental data, unless the reviewer feels strongly about this.

Other important issues

Line 184. “Myosin enrichment at the lateral cortex”. It needs to be explained precisely what they mean here by "Myosin enrichment on the lateral cortex". Enriched compared to what? It is already enriched at the lateral cortex compared to the apical pole at 0s, it's enriched at the lateral pole at 30s vs 0s, and it's already enriched at the lateral pole compared to the basal pole at 60s.

To provide some clarity, we measured this in high temporal resolution movies and explained it better in Figure 1.

Measurements for the delay should be shown for *sds22* to demonstrate it's not involved in clearing.

This data is provided in supplemental Figure 7f.

Figure 2c has to be rod+colc, not colcemid as panel and legend suggest. Without rod clearing of basal Myosin cannot occur.

Thank you for pointing this out; the figure is modified accordingly.

Fig 5c – how can you possibly determine anaphase in rod colc neuroblasts?

We have measured apical clearing time in relation to anaphase onset in wild type and since apical clearing occurs normally in colcemid treated *rod* mutant neuroblasts, we are confident that these neuroblasts are in anaphase.

Line 244: On what basis is apical determined in *dlg;pins* (check nomenclature) neuroblasts that express Myosin? larger = apical? This would be a circular argument.

Although we are aware that *dlg;pins* mutants are unpolarized, we refer to the pole of the neuroblast giving rise to the slightly bigger cell as “apical” and the pole of the cell giving rise to the smaller cell as “basal”. We apply this nomenclature because we assume that if the cortex clears earlier or expands more, it is more comparable to the “wild type” apical cortex. Also, the goal of these measurements was to show that the expansion of both poles are very similar in *dlg;pins* neuroblasts; both poles expand more than the basal wild type cortex. In order to not over-estimate this difference, we compared the expansion of the basal wt pole with the *dlg;pins* pole displaying the “smallest” expansion. We chose this secure way knowing that in the worst case, the difference will be a bit under-estimated (which has no consequence on the conclusion presented here).

Given the nature of the dataset used for Myosin speed measurements their value is questionable.

We have performed FRAP experiments and measured flow velocity in kymographs. The resulting values are very similar to the ones obtained from our photoconversion experiments.

There is a typo in the title.

Thank you for pointing this out. We corrected the title.

Line 287 should refer to Fig7.

Line 416 check color annotation.

Figure numbering changed but we hope we corrected all annotations and figure callings.

Reviewers' Comments:

Reviewer #1:

Remarks to the Author:

The authors have done an good job addressing my previous concerns, and I am happy to support publication. However, there is one point that I believe still deserves attention:

I still find myself unconvinced of the conclusions drawn from analysis of the FRET biosensor for Myosin activity presented in Figure 5. The authors are claiming that activity increases at the equator before basal clearing, but based on what exactly? The representative data shown in Figure 5d suggests that the FRET data for one cell is very noisy and raises concerns about how one would determine the onset of a rise in FRET activity. So far as I can tell, the authors provide no info about how this determination was made. It is not clear from the scatter plots in 5e and 5g that there is a significant n the onset of FRET activity vs basal clearing and so far as I can tell, the authors present no analysis of this difference, beyond a verbal claim that it is different. I think these data and the analysis upon which their conclusions are based must be presented with more care.

Reviewer #2:

Remarks to the Author:

The authors have conducted a series of new experiments to address the points raised by the referees. Especially, the flavopiridol experiments and ROCK kinase apical targeting strategy have yielded important new insights about the temporal and spatial control of events during asymmetric cell division. Based on this and although the authors have not used myosin inhibition directly, I am supportive of the publication of the revised version of the manuscript.

The authors should ensure that the wording of impact of flows on phenotypic effects throughout the manuscript is appropriate and not exaggerated.

Reviewer #3:

Remarks to the Author:

The revised manuscript is greatly improved over the previous version. It is a timely and relevant topic. However, there remain issues that need attention before publication.

The correlation between clearing delay and physical asymmetry in control neuroblasts is very convincing (Fig S7a). It should be given more emphasis in the text and should be part of the main figures. Fig 7c should also be updated with these additional observations.

The authors describe the mechanism clearing Myosin from the apical pole as a "basally directed Myosin flow originating on the apical cortex". They propose two arguments in favor of this proposition: 1) photoconverted Myosin at the lateral furrow after apical clearing does not relocalizes to the entire furrow, but rather stays on the site where it was initially photoconverted (Fig 2d). 2) After photobleaching at the lateral cortex after apical clearing, Myosin fills in the bleached region from both the apical and basal edges (Fig 2e-f). However, neither of these experiments allows the authors to claim a flow originates at the apical pole: Both are performed at the lateral cortex, after the apical clearing of Myosin. In other words, at the wrong place and at the wrong time.

Former figure 6d (now 2d) needs to include the nonconverted myosin channel to know when in relation to onset of flows conversion was carried out, if again it is after apical clearing, then the

issue remains open. Why don't the flows originate laterally?

Myosin recruitment in metaphase and anaphase cuts are consistent with the idea that the cortex progressively loses competence to receive myosin. Therefore it needs to be emphasized in the manuscript that local downregulation of Myosin at the apical pole cannot be excluded. It might be good to temper the conclusion about where the flows originate (remove from abstract) since this appears to be (as actually in a few cases) subject of another paper in revision.

Moreover, how does "Myosin coming from both the apical and basal edges" after FRAP confirm the existence of directed flows? The metaphase control is interesting, but a better control might be a "naïve" membrane marker like Gap43 in anaphase. Ideally, since the basal-to-apical flow starts 60s after the basal flow, there should be a 60 seconds time window during which only the apical-to-basal flow should contribute to Myosin recovery, thus one would expect Myosin to only fill the gap from the apical side. I appreciate that this is tricky, but the paper is about the precise spatial temporal regulation asymmetric kymograph profiles would be very convincing.

Regarding this, when exactly was the photobleaching experiment shown Fig 2e-f performed? No basal clearing is seen, doesn't this suggest the basal flow actually hasn't started yet? So why then is the kymograph symmetrically recovering from the apical and basal side?

L275, "the increase in activated Myosin could be the motor for basal Myosin clearing" based on: "both Myosin and F-Actin intensity dropped on the basal neuroblast cortex shortly before furrowing initiated" and Myosin and F-Actin intensity dropped "after activated Myosin was detected in the prospective furrow region". However, Fig 5g clearly shows that basal Myosin and F-Actin intensity can also drop after furrowing initiation. More importantly, comparing Fig 5e and Fig 5g shows that the drop of basal F-Actin and Myosin (occurring earliest 60s before furrowing) can occur before the activation of Myosin at the lateral cortex (occurring earliest 40s before furrow initiation). Doesn't this contradict the proposition that "the increase in activated Myosin could be the motor for basal Myosin clearing"?

The manuscript now includes an interesting set of experiments using nanobodies. For instance, L329, the authors test how Myosin relocalization dynamics influence sibling cell size asymmetry by forcing Sqh::GFP localization to the apical pole in neuroblasts depleted for endogenous sqh, which they claim results in the inversion of the physical asymmetry between daughter cells (Fig 7d). This is a great result.

However, they express ALD-Rockca::mCherry (did I get that right Rock or Rockca? check ROCK annotation in manuscript and figures) alone and ALD-Rockca::nanobody fusion, so the obvious control expressing ALD-nanobody in a Sqh::GFP ; sqh background is missing. Do we need the Rockca there, or is manipulating Myosin mobility apically already enough? That is important to distinguish. You can not be sure that only myosin distribution and activity are altered. What happens to actin here?

Moreover they compare ALD-Rockca::nanobody; Sqh::GFP; sqh to wild type in figure 7. This is definitely not the right control for that experiment. ALD-Rockca::nanobody; your favourite-GFP; or ALD-Rockca::mCherry; Sqh::GFP; sqh must be used.

One obvious concern with this experiment is that size effects do not result from defective Myosin apical clearing, but rather from defective polarity, which the authors address by showing that relocalizing Sqh::GFP apically does not affect a basal marker (Fig S7d). However, The legend of Fig S7d does not state whether neuroblasts are mutant for sqh, as they were for Fig 7d. If not, this does not represent a valid control. I was prompted to this since the Sqh::GFP localization shown in Fig 7d (enriched apically but present all over the cortex) looks very different from Fig S7d (exclusively apical). So the difference in genotype seems to be more likely and perhaps endogenous Sqh, then rescues polarity? An anaphase cell (of genotype as in fig 7d) stained for an apical and basal marker needs to be provided (small cell with apical marker and larger cell with basal marker).

Importantly, assuming that polarity is not affected and given the UAS-driven nature of the tools

used (no temporal/conditional control mentioned), I am utterly confused, why large neuroblasts are obtainable. These NBs have been dividing for many rounds already and this effect should occur at each division (the quantification shows quite robust effects, making NBs either divide symmetric or even asymmetric yielding smaller "apical" cells). So why are there still >10 μ m diameter neuroblasts in Two rounds of division need to be shown. So while the nanobody set of experiments is potentially interesting it raises more concerns than it helps provide arguments for their model and would need a great deal of additional controls to be acceptable.

Reviewers' comments:

Reviewer #1 (Remarks to the Author):

The authors have done an good job addressing my previous concerns, and I am happy to support publication. However, there is one point that I believe still deserves attention:

I still find myself unconvinced of the conclusions drawn from analysis of the FRET biosensor for Myosin activity presented in Figure 5. The authors are claiming that activity increases at the equator before basal clearing, but based on what exactly? The representative data shown in Figure 5d suggests that the FRET data for one cell is very noisy and raises concerns about how one would determine the onset of a rise in FRET activity. So far as I can tell, the authors provide no info about how this determination was made. It is not clear from the scatter plots in 5e and 5g that there is a significant n the onset of FRET activity vs basal clearing and so far as I can tell, the authors present no analysis of this difference, beyond a verbal claim that it is different. I think these data and the analysis upon which their conclusions are based must be presented with more care.

We have reanalyzed the data provided in Figure 5. To accurately identify the onset of FRET activity, we first smoothed the curve with the 'loess' method using a smoothing factor of 0.4 to reduce local fluctuations. We then determined the local minimum that lies closest to the left of the monotonically rising smoothed curve as the signal increases. The zero gradient value is used as a criterion to identify local minima/maxima. For the onset of basal Myosin and Actin intensity drop, the local maximum that lies closest to the left of the monotonically decreasing smoothed curve are used.

Finally, we also included statistical values to test for significance.

Based on this new data, we found that the increase in Myosin activity in the future furrow region occurs at the same time as the relocalization of basal Myosin and F-Actin. We think this finding does not necessarily argue against our model but shows that lateral Myosin activity increase can coincide with basal clearing. Unfortunately we are unable to simultaneously image the FRET sensor in conjunction with Myosin and/or LifeAct. Thus, we cannot directly compare these measurements from individual cells but have to compare two independent datasets.

We included an improved version of Figure 5 into a revised manuscript and also modified the wording in the manuscript.

Reviewer #2 (Remarks to the Author):

The authors have conducted a series of new experiments to address the points raised by the referees. Especially, the flavopiridol experiments and ROCK kinase apical targeting strategy have yielded important new insights about the temporal and spatial control of events during asymmetric cell division. Based on this and although the authors have not used myosin inhibition directly, I am supportive of the publication of the revised version of the manuscript.

The authors should ensure that the wording of impact of flows on phenotypic effects throughout the manuscript is appropriate and not exaggerated.

We modified the text wherever necessary

Reviewer #3 (Remarks to the Author):

The revised manuscript is greatly improved over the previous version. It is a timely and relevant topic. However, there remain issues that need attention before publication.

The correlation between clearing delay and physical asymmetry in control neuroblasts is very convincing (Fig S7a). It should be given more emphasis in the text and should be part of the main figures. Fig 7c should also be updated with these additional observations.

We combined the wild type measurements, performed on 50 cells, with the other conditions in one graph shown in Figure 7c.

The authors describe the mechanism clearing Myosin from the apical pole as a “basally directed Myosin flow originating on the apical cortex”. They propose two arguments in favor of this proposition: 1) photoconverted Myosin at the lateral furrow after apical clearing does not relocalizes to the entire furrow, but rather stays on the site where it was initially photoconverted (Fig 2d). 2) After photobleaching at the lateral cortex after apical clearing, Myosin fills in the bleached region from both the apical and basal edges (Fig 2e-f). However, neither of these experiments allows the authors to claim a flow originates at the apical pole: Both are performed at the lateral cortex, after the apical clearing of Myosin. In other words, at the wrong place and at the wrong time.

The photoconversion experiment displayed in Figure 2d - now also showing the unconverted channel - very clearly indicates that photoconversion was performed on the apical cortex. So, although we can say that apical Myosin flows towards the basal cortex, the exact origin of the flows might remain elusive. Thus, we reworded the manuscript wherever necessary.

Former figure 6d (now 2d) needs to include the nonconverted myosin channel to know when in relation to onset of flows conversion was carried out, if again it is after apical clearing, then the issue remains open. Why don't the flows originate laterally?

We replaced the panel in Figure 2d with this new panel here.

We deliberately performed photoconversion right after the onset of apical clearing to catch the flow phase. It is technically impossible to do the photoconversion exactly at the time point when the flow starts. Also, if photoconversion is performed too early, there is still an exchange between cortical and cytoplasmic Myosin occurring; Myosin will redistribute over the entire cortex.

Figure 2

Myosin recruitment in metaphase and anaphase cuts are consistent with the idea that the cortex progressively loses competence to receive myosin. Therefore it needs to be emphasized in the manuscript that local downregulation of Myosin at the apical pole cannot be excluded. It might be good to temper the conclusion about where the flows originate (remove from abstract) since this appears to be (as actually in a few cases) subject of another paper in revision.

We modified the text accordingly, mentioning the possibility of local Myosin – both apical and basal - activity downregulation.

Moreover, how does "Myosin coming from both the apical and basal edges" after FRAP confirm the existence of directed flows? The metaphase control is interesting, but a better control might be a "naïve" membrane marker like Gap43 in anaphase.

I would like to point out that we have already provided a control on a photoconvertible membrane marker; as the data in Supplemental Figure 2h clearly shows, laterally photoconverted Gap43 (membrane marker) spreads out over the entire cortex, which is completely different from Myosin's behavior.

We also performed FRAP experiments on neuroblasts expressing Sqh::GFP together with membrane tethered mCherry (mCherry::CAAX; supplemental Figure 2f-i) or Sqh::mCherry together with PH::GFP (not shown). Both membrane markers recovered with completely different dynamics than Sqh::GFP or Sqh::mCherry.

Ideally, since the basal-to-apical flow starts 60s after the basal flow, there should be a 60 seconds time window during which only the apical-to-basal flow should contribute to Myosin recovery, thus one would expect Myosin to only fill the gap from the apical side. I appreciate that this is tricky, but the paper is about the precise spatial temporal regulation asymmetric kymograph profiles would be very convincing.

Regarding this, when exactly was the photobleaching experiment shown Fig 2e-f performed? No basal clearing is seen, doesn't this suggest the basal flow actually hasn't started yet? So why then is the kymograph symmetrically recovering from the apical and basal side?

The experiment the reviewer proposes here is very challenging. Our data do not argue against a basally and an apically directed Myosin flow. The manifestation of Myosin clearing on the apical and/or basal flow simply indicates that the flows already started. Thus, the FRAP experiment does not exclude the possibility that the flows are already underway at the position where Myosin was FRAPed. It just did not yet cause basal Myosin to clear.

Taken together, we have performed photoconversion experiments on a photoconvertible membrane marker, FRAP experiments at metaphase and on a membrane tethered fluorescent protein. In all cases, the relocalization dynamics starkly differed from Myosin's. Based on all this data, we are confident that Myosin flows occur with the described spatiotemporal dynamics.

L275, "the increase in activated Myosin could be the motor for basal Myosin clearing" based on: "both Myosin and F-Actin intensity dropped on the basal neuroblast cortex shortly before furrowing initiated" and Myosin and F-Actin intensity dropped "after activated Myosin was detected in the prospective furrow region". However, Fig 5g clearly shows that basal Myosin and F-Actin intensity can also drop after furrowing initiation. More importantly, comparing Fig 5e and Fig 5g shows that the drop of basal F-Actin and Myosin (occurring earliest 60s before furrowing) can occur before the activation of Myosin at the lateral cortex (occurring earliest 40s before furrow initiation). Doesn't this contradict the proposition that "the increase in activated Myosin could be the motor for basal Myosin clearing"?

As mentioned above, we reanalyzed the data shown in Figure 5. In almost all investigated cases we found that increase in activated Myosin precedes furrowing. Similarly, in most cases, Myosin and F-Actin intensity dropped on the basal cortex prior to furrowing. Unfortunately, we are unable to simultaneously image the FRET sensor in conjunction with Myosin and/or LifeAct. Thus, we cannot directly compare these measurements from individual cells but compare two independent datasets.

We included an improved version of Figure 5 into a revised manuscript and also modified the wording in the manuscript.

The manuscript now includes an interesting set of experiments using nanobodies. For instance, L329, the authors test how Myosin relocalization dynamics influence sibling cell size asymmetry by forcing Sqh::GFP localization to the apical pole in neuroblasts depleted for endogenous sqh, which they claim results in the inversion of the physical asymmetry between daughter cells (Fig 7d). This is a great result.

However, they express ALD-Rockca::mCherry (did I get that right Rock or Rockca? check ROCK annotation in manuscript and figures) alone and ALD-Rockca::nanobody fusion, so the obvious control expressing ALD-nanobody in a Sqh::GFP ; sqh background is missing. Do we need the Rockca there, or is manipulating Myosin mobility apically already enough? That is an important one to distinguish. You can not be sure that only myosin distribution and activity are altered. What happens to actin here?

I agree with the reviewer that this is a good point. However, we have another manuscript in preparation, showing that Myosin phosphorylation – using a similar nanobody – is sufficient to alter Myosin’s localization, but the same nanobody containing a kinase dead version of Rock’s kinase domain is unable to relocalize Sqh::GFP. Based on these experiments, we suspect that both the nanobody and the kinase domain are required to alter Myosin’s localization. Apical localized Rock^{CA} could fail to recapitulate the nanobody experiment due to lack of proximity, incompatible protein topology or other reasons.

For time reasons, we have not generated an ALD-nanobody line but modified the manuscript, mentioning that the nanobody could either trap Myosin apically, induce inverted asymmetry due to locally activated Myosin or a combination of both.

Since Myosin is the motor protein, exerting forces onto Actin filaments, our focus was to show that manipulating Myosin is sufficient to alter physical asymmetry. I don’t think it is necessary to also show F-Actin distribution especially since manipulating Myosin is sufficient to invert physical asymmetry.

Moreover they compare ALD-Rockca::nanobody; Sqh::GFP; sqh to wild type in figure 7. This is definitively not the right control for that experiment. ALD-Rockca::nanobody; your favourite-GFP; or ALD-Rockca::mCherry; Sqh::GFP; sqh must be used.

Anne Royou has shown that Sqh::GFP can rescue the sqh mutant phenotype. The purpose to use sqh mutant cells – rescued by Sqh::GFP – is to replace the endogenous Sqh protein pool with GFP-tagged Sqh, so it can be manipulated with the nanobody.

sqh mutant neuroblasts expressing Sqh::GFP are indistinguishable from wild type neuroblasts expressing Sqh::GFP in terms of cortical expansion. Thus, we considered it fair to compare cortical expansion of “normally” behaving neuroblasts with nanobody expressing neuroblasts.

One obvious concern with this experiment is that size effects do not result from defective Myosin apical clearing, but rather from defective polarity, which the authors address by showing that relocalizing Sqh::GFP apically does not affect a basal marker (Fig S7d).

However, The legend of Fig S7d does not state whether neuroblasts are mutant for sqh, as they were for Fig 7d. If not, this does not represent a valid control. I was prompted to this since the Sqh::GFP localization shown in Fig 7d (enriched apically but present all over the cortex) looks very different from Fig S7d (exclusively apical). So the difference in genotype seems to be more likely and perhaps endogenous Sqh, then rescues polarity?

We stained *sqh* mutant neuroblasts, expressing rock^{CA}-vhhGFP4 together with Sqh::GFP, with aPKC or Miranda. Miranda was normally localized in metaphase neuroblasts (with the aforementioned genotype). Since basal Mira localization depends on the correct localization of aPKC, which is dependent on the apical Par complex, we conclude that neuroblast polarity is normally established in *sqh* mutant neuroblasts, expressing the functionalized nanobody together with Sqh::GFP

In support of this data, we also found anaphase neuroblasts (again, with the aforementioned genotype) with polarized aPKC; enriched Myosin was colocalizing with aPKC, strongly suggesting that this corresponds to the apical neuroblast cortex.

An anaphase cell (of genotype as in fig 7d) stained for an apical and basal marker needs to be provided (small cell with apical marker and larger cell with basal marker).

Despite several attempts, we could only find an intermediate stage, showing identically sized sibling cells; the sibling cell containing stronger Sqh::GFP signal was also positive for aPKC. It is very difficult to catch neuroblasts at the correct stage, especially since the nanobody induced reversion phenotype is predominantly found in older brains, which don't show as many dividing neuroblasts as younger brains, thereby reducing the probability of finding these cells in fixed tissues.

Importantly, assuming that polarity is not affected and given the UAS-driven nature of the tools used (no temporal/conditional control mentioned), I am utterly confused, why large neuroblasts are obtainable. These NBs have been dividing for many rounds already and this effect should occur at each division (the quantification shows quite robust effects, making NBs either divide symmetric or even asymmetric yielding smaller "apical" cells). So why are there still >10 μ m diameter neuroblasts in Two rounds of division need to be shown.

We have not observed this phenotype in young but only in old third instar larvae, explaining why we still find neuroblasts with >10 μ m diameter. Thus, we do not think that the phenotype is already manifested in every division and preferentially selected older larvae for our live cell imaging experiments.

So while the nanobody set of experiments is potentially interesting it raises more concerns than it helps provide arguments for their model and would need a great deal of additional controls to be acceptable.

Reviewers' Comments:

Reviewer #1:

Remarks to the Author:

The author's have addressed my issue with the FRET analysis. Overall, this paper provides valuable new insights and I support its publication without reservation. One suggestion: The authors should use a term like "increased FRET signal" rather than "myosin activity" or "active tension" in describing their observations to more clearly distinguish between the observation and its interpretation...

Reviewer #3:

Remarks to the Author:

The first part of the manuscript has improved and is of general interest for cell and developmental biology. Some of the data has been re-analysed and now better fit the conclusions, which has cleared up confusion in the manuscript. The main conclusions are supported by data most of which are convincing (I am still doubtful about the meaningfulness of the FRAP data and struggle to see why when photo-conversion at specific time points is possible, FRAP is not). Overall, I would be supportive of this part being published, however. It would be a big plus that the exact genotypes of the experiments shown appear in the figure legends.

The nanobody experiments, in contrast, remain incomplete and their interpretation is much more complex as currently presented. The rationale of the 'functionalized nanobody' is not properly introduced making this set of experiments hard to follow (after reading the rebuttal, it is unclear to me if this nanobody can potentially redirect the localization of any GFP tagged protein?). I was further under the assumption that the ROCKCA nanobody always (not quantified, however) mislocalizes Sqh::GFP apically in neuroblasts. It now appears that the phenotype (size inversion) is only occurring in old larvae (this crucial information is missing in the manuscript).

We do not know how this relates to the apical enrichment of Myosin (larval age is not a criterion in the relevant data shown). Therefore, the interpretation that locally activated Myosin biases the flows is not so simple (How to rule out accumulated damage or an age-specific, therefore less general effect?). There is no VIDEO showing two consecutive divisions, which might have been helpful here. I pointed this out previously, which unfortunately went uncommented in the rebuttal. Sqh::GFP in a sqh mutant background (confusingly called wild type) continues not to be the right control for Fig 7f. The problem is not Sqh::GFP functionality. The effect of expression of ALD-ROCKCA on the expansions that are measured needs to be controlled for (e.g. ALD-ROCKCA-mCherry in Sqh::GFP sqh mutant background as previously suggested).

Reviewers' comments:

Reviewer #1 (Remarks to the Author):

The author's have addressed my issue with the FRET analysis. Overall, this paper provides valuable new insights and I support its publication without reservation. One suggestion: The authors should use a term like "increased FRET signal" rather than "myosin activity" or "active tension" in describing their observations to more clearly distinguish between the observation and its interpretation...

We have changed the manuscript accordingly.

Reviewer #3 (Remarks to the Author):

The first part of the manuscript has improved and is of general interest for cell and developmental biology. Some of the data has been re-analysed and now better fit the conclusions, which has cleared up confusion in the manuscript. The main conclusions are supported by data most of which are convincing (I am still doubtful about the meaningfulness of the FRAP data and struggle to see why when photo-conversion at specific time points is possible, FRAP is not). Overall, I would be supportive of this part being published, however.

It would be a big plus that the exact genotypes of the experiments shown appear in the figure legends.

We have modified the legends wherever necessary, indicating the exact genotype for each experiments. In most instances, the genotype is already indicated in the figures.

The nanobody experiments, in contrast, remain incomplete and their interpretation is much more complex as currently presented. The rationale of the 'functionalized nanobody' is not properly introduced making this set of experiments hard to follow (after reading the rebuttal, it is unclear to me if this nanobody can potentially redirect the localization of any GFP tagged protein?).

We are surprised by this critique since in the previous revisions, reviewer # 3 accurately summarized the rationale of the functionalized nanobody: "... the authors test how Myosin relocalization dynamics influence sibling cell size asymmetry by forcing Sqh::GFP localization to the apical pole in neuroblasts depleted for endogenous sqh, which they claim results in the inversion of the physical asymmetry between daughter cells (Fig 7d). This is a great result.

Needless to say, we cannot prove that the nanobody is able to redirect the localization of ANY GFP tagged protein. [redacted]

(1) [redacted]

[Redacted]

[Redacted]

[Redacted]

Furthermore, we strongly believe that the data currently presented in the manuscript are sufficient to interpret the experiment: in contrast to an apically localized kinase domain of Rock, only the nanobody functionalized with RockCA and tethered to the apical neuroblast cortex is able to change Myosin dynamics and to invert physical asymmetry (see also below). Although we have shown that the nanobody increases the amount of phosphorylated Myosin on the apical cortex, we modified the manuscript in the previous round of revisions by stating that either local increase in phosphorylated Myosin or apical trapping of Myosin is causing an inversion of physical asymmetry.

I was further under the assumption that the ROCKCA nanobody always (not quantified, however) mislocalizes Sqh::GFP apically in neuroblasts. It now appears that the phenotype (size inversion) is only occurring in old larvae (this crucial information is missing in the manuscript).

We have not further investigated why size inversion only occurs in old larvae. One reason could be maternal contribution; mothers, carrying one wild type allele of Sqh might provide enough wild type Sqh protein to rescue the nanobody phenotype until it completely dilutes out. For the hypothesis to be tested and for the overall topic of the manuscript, it is irrelevant when the phenotype occurs. The dynamics of the phenotype is depending on several factors (Gal4 expression dynamics, nanobody accumulation etc), many of which cannot be easily controlled or quantified.

We have modified the manuscript, indicating that the phenotype predominantly occurs in old larvae.

We do not know how this relates to the apical enrichment of Myosin (larval age is not a criterion in the relevant data shown). Therefore, the interpretation that locally activated Myosin biases the flows is not so simple (How to rule out accumulated damage or an age-specific, therefore less general effect?).

We are of the opinion that our data already address the reviewers concern:

- (1) We show that only nanobody expressing neuroblasts retain apical Myosin (Figure 7f,g) and also accumulation of activated Sqh1P (Supplemental figure 7c). Thus, we do know how nanobody expression relates to apical enrichment of Myosin.
- (2) We also show that only nanobody expressing neuroblasts affect apical expansion, while allowing basal expansion, arguing for the spatial restricted (apical) activity of the nanobody (Figure 7h and Supplemental Figure 7a,b).
- (3) It is unlikely that accumulated damage or other less general effects would cause the inverted phenotype since nanobody expressing neuroblasts are normally polarized and seem to divide as fast as wild type neuroblasts.

However, to strengthen our conclusions, we quantified apical Myosin enrichment in *sqh* mutant neuroblasts expressing ALD-rockCA::vhhGFP4 and Sqh::GFP and compared it to wild type neuroblasts expressing Sqh::GFP only. To this end, we measured apical and basal Myosin on kymographs during anaphase and calculated the ratios (apical/basal) of averaged Myosin intensity. This data shows, that in contrast to wild type (Sqh::GFP expression only) neuroblasts, neuroblasts expressing ALD-rockCA::vhhGFP4 and Sqh::GFP retain higher amounts of apical Myosin during anaphase. In other words, whereas apical clearing occurs during wildtype anaphase neuroblasts, Myosin remains apically localized on the apical neuroblast cortex in nanobody expressing neuroblasts. This data is supporting our previous conclusions, showing that nanobody expressing neuroblasts alter apical Myosin dynamics manifested in (1) an increase in Sqh1P labelled Myosin and (2) an increase in Sqh::GFP on the apical cortex. This new data is shown in Figure 7h.

There is no VIDEO showing two consecutive divisions, which might have been helpful here. I pointed this out previously, which unfortunately went uncommented in the rebuttal.

We apologize for not commenting on this earlier. We have not shown two consecutive division for technical and conceptual reasons:

- (1) to acquire data with relatively high temporal resolution, our imaging settings were optimized to acquire data with relatively high temporal resolution. However, to avoid bleaching and photodamage, we stopped these acquisitions after neuroblasts completed cytokinesis. To show two consecutive divisions, movies with lower temporal resolution would have to be acquired to prevent bleaching and photodamage (a neuroblast cell cycle can take more than 2h in older larvae).
- (2) While it would be useful and nice to show two consecutive divisions, this experiment would not address the hypothesis we set out to test but would be addressing the question how inverted polarity affects cell fate and behavior. Needless to say, this is not the topic of this manuscript. As outlined above, at the time of physical asymmetry inversion, the neuroblasts are properly polarized, are similar in size to wild type neuroblasts and do not show any obvious defects or phenotypes other than the subsequent size inversion. Thus, we are confident to state that these cells are of neuroblast fate, switching their physical asymmetry. What will happen in the following division is not the topic of this manuscript but will be addressed in subsequent studies.

Sqh::GFP in a *sqh* mutant background (confusingly called wild type) continues not to be the right control for Fig 7f. The problem is not Sqh::GFP functionality. The effect of expression of ALD-ROCKCA on the expansions that are measured needs to be controlled for (e.g. ALD-ROCKCA-mCherry in Sqh::GFP *sqh* mutant background as previously suggested).

RockCA::mCherry does not change the localization of Sqh::GFP in interphase S2 cells. More importantly, ALD-RockCA::mCherry does also not change cortical expansion nor physical asymmetry in neuroblasts. Although the reasons for this are not entirely clear (maybe due to physical proximity of the kinase domain to the substrate, steric hindrance or other reasons...), the phenotype is very clear: ALD-RockCA::mCherry expressing neuroblasts look like wild type. Thus, comparing neuroblasts expressing Sqh::GFP with or without ALD-RockCA::mCherry does not make a difference and for ease of use, we decided to use neuroblasts expressing only Sqh::GFP.

Reviewers' Comments:

Reviewer #3:

Remarks to the Author:

The key observations reported in this revised manuscript are intriguing and very interesting. Likewise, the concept of the functionalized nanobody and the results observed with it are potentially interesting. However, under a little scrutiny the interpretation of the results obtained with the nanobody based tools proved as the authors admit to be much more complex than initially presented relativizing their impact. The authors decided to argue away suggested simple experiments that might have made this part stronger. As I intended to suggest already, I think this set of experiments is not necessary to convey the main message. The paper would be robust enough for publication and certainly less bulky without it. The description of the flows is great and for most parts convincingly documented and the proposed concepts are interesting. I have voiced my concerns, so I leave the decision to publish the part containing the nanobody results, which I still think raises more questions than it answers, with the authors.

REVIEWERS' COMMENTS:

Reviewer #3 (Remarks to the Author):

The key observations reported in this revised manuscript are intriguing and very interesting. Likewise, the concept of the functionalized nanobody and the results observed with it are potentially interesting. However, under a little scrutiny the interpretation of the results obtained with the nanobody based tools proved as the authors admit to be much more complex than initially presented relativizing their impact.

We are convinced that the provided data is sufficient to interpret the results and to support our model. However, upon the reviewer's suggestion, we mentioned the complexities with the nanobody tool in the manuscript.

The authors decided to argue away suggested simple experiments that might have made this part stronger.

Based on the reviewer's suggestion, we performed several additional experiments and quantification – all of which strengthened our conclusions and model. We have not tried to simply argue away suggested experiments but performed the ones which we deemed reasonable and technically feasible.

As I intended to suggest already, I think this set of experiments is not necessary to convey the main message. The paper would be robust enough for publication and certainly less bulky without it. The description of the flows is great and for most parts convincingly documented and the proposed concepts are interesting. I have voiced my concerns, so I leave the decision to publish the part containing the nanobody results, which I still think raises more questions than it answers, with the authors.

We would like to thank reviewer 3 for his thoughtful comments and critiques. However, we consider these experiments a very valuable addition to the manuscript, strengthening our main conclusions and model. For these reasons, we decided to leave the nanobody data in the manuscript.